# Applying machine learning to improve the near-real-time products of the Aura Microwave Limb Sounder

Frank Werner[1], Nathaniel J. Livesey[1], Luis F. Millán[1], William G. Read[1], Michael J. Schwartz[1], Paul A. Wagner[1], William H. Daffer[1], Alyn Lambert[1], Sasha N. Tolstoff[2], and Michelle L. Santee[1]

[1]Jet Propulsion Laboratory, California Institute of Technology, 4800 Oak Grove Drive, Pasadena, CA 91109, USA
[2]California Institute of Technology, 1200 East California Blvd, Pasadena, CA 91105, USA

**Correspondence:** Frank Werner (frank.werner@jpl.nasa.gov)

**Abstract.** A new algorithm to derive near-real-time (NRT) data products for the Aura Microwave Limb Sounder (MLS) is presented. The old approach was based on a simplified optimal estimation retrieval algorithm (OE-NRT) to reduce computational demands and latency. This manuscript describes the setup, training, and evaluation of a redesigned approach based on artificial neural networks (ANN-NRT), which is trained on $> 17$ years of MLS radiance observations and composition profile retrievals. Comparisons of joint histograms and performance metrics derived between the two NRT results and the operational MLS products demonstrate a noticeable statistical improvement from ANN-NRT. This new approach results in higher correlation coefficients, as well as lower root-mean-square deviations and biases at almost all retrieval levels compared to OE-NRT. The exceptions are pressure levels with concentrations close to $0$ ppbv, where the ANN models fail to establish a functional relationship and tend to predict zero. Depending on the application, this behavior might be advantageous. While the developed models can take advantage of the extended MLS data record, this study demonstrates that training ANN-NRT on just a single year of MLS observations is sufficient to improve upon OE-NRT. This confirms the potential of applying machine learning to the NRT efforts of other current and future mission concepts.

## 1 Introduction

The Aura Microwave Limb Sounder (MLS) data record is more than $18$ years long, far exceeding the MLS 5-year design life. Due to its exceptionally long duration and reliability (e.g., Hubert et al., 2016; Hegglin et al., 2021; Read et al., 2022), MLS observations are employed to study a wide range of atmospheric science topics, such as long-term trends in atmospheric constituents (e.g., Gaudel et al., 2018; Lossow et al., 2018; Strahan and Douglass, 2018; Froidevaux et al., 2019), global troposphere-stratosphere transport (e.g., Neu et al., 2014; Diallo et al., 2019), the influence of strong convective systems on lower-stratospheric humidity (e.g., Schwartz et al., 2013; Werner et al., 2020), as well as the impact of wildfires and volcanic eruptions on stratospheric chemistry (e.g., Pumphrey et al., 2015; Schwartz et al., 2020; Millán et al., 2022; Santee et al., 2022), to name just a few.

Processing of the standard retrieval products provided by MLS takes a little less than a full day and thus cannot be used in near-real-time (NRT) applications. Therefore, the MLS team started providing NRT data based on a simplified retrieval algorithm for a limited selection of its standard species in 2008. These products are routinely produced within 3 hours of the MLS observations (Lambert et al., 2022) and can thus be delivered to the scientific community much more expeditiously. Examples of MLS NRT usage are the assimilation of MLS NRT ozone ($O_3$) profiles into the Copernicus Atmosphere Monitoring Service (CAMS) from the European Centre for Medium-Range Weather Forecasts (ECMWF) (e.g., Peuch et al., 2022), as well as deliveries of $O_3$, water vapor ($H_2O$), and carbon monoxide (CO) maps over Southeast Asia during the Asian Summer Monsoon Chemical & Climate Impact Project (ACCLIP; https://www.eol.ucar.edu/field_projects/acclip/, last access: 19 December 2022) campaign in 2022 (Pan et al., 2022). MLS NRT $O_3$ and temperature ($T$) profiles are also assimilated by the numerical weather prediction model of the Naval Research Laboratory (Hoppel et al., 2008), while NRT $H_2O$ and sulphur dioxide ($SO_2$) are part of the NASA Major Volcanic Eruption Response Plan (NASA, 2018). While MLS NRT data help to constrain the model forecasts, monitor the stratopshere during volcanic eruptions, and aid flight planning during aircraft campaigns, they are less reliable than the standard MLS products and require careful screening procedures (Lambert et al., 2022).

Recent years have seen a proliferation of the application of machine learning approaches in atmospheric sciences, from dimensionality reduction of satellite observations (e.g., Del Frate et al., 2005), to estimates of aerosol particle loading (e.g., Grivas and Chaloulakou, 2006) and cloud cover (e.g., Saponaro et al., 2013; Werner et al., 2020), to land cover studies (e.g., Campos-Taberner et al., 2020), to weather and climate modelling (e.g., Schultz et al., 2021). Two of the main benefits of applying machine learning techniques to answer atmospheric science questions are (i) pattern recognition enabling identification of previously unknown or poorly understood relationships between observations and the atmospheric state and (ii) the increase in computational efficiency leading to faster turnaround times in predicting the atmospheric variable of interest.

In this study we describe an updated Aura MLS NRT setup that applies artificial neural networks (ANN) to facilitate faster and more reliable predictions of MLS NRT constituent profiles. This new algorithm provides both of the above mentioned benefits of machine learning techniques: (i) it pinpoints the relevant MLS radiance observations that reliably determine the individual species profiles and (ii) yields NRT profile predictions an order of magnitude faster than the previous algorithm it replaces. The manuscript is structured as follows: an introduction to MLS observations, retrieved data products, and retrieval algorithms is given in section 2. An overview of the ANN setup, training, and evaluation is presented in section 3. A comparison of the former and updated NRT algorithm encompassing joint histograms, performance metrics, and global maps is given in section 4. The main conclusions and a brief summary are presented in section 6.

## 2   Data

Aura MLS has observed brightness temperatures from five spectral frequency ranges centered around 118, 190, 240, 640, and 2,500 GHz since 2004 (Waters et al., 2006). The 2,500 GHz band targeted the hydroxyl radical; it was deactivated in 2010 and is not considered here. Table 4 in Waters et al. (2006) and Figure 2.1.1 in Livesey et al. (2022) give an overview and additional details on individual MLS bands and channels as well as the specific absorption characteristics of the various atmospheric

constituents that are targeted. Daily MLS observations comprise $\approx 3500$ vertical limb scans (called major frames; MAFs), each of which takes $\approx 20\,\mathrm{s}$ to complete. Each MAF consists of 125 radiance integrations (called minor frames; MIFs) during a continuous vertical scan of the limb. In this study, MLS brightness temperatures sampled over 2005–2022 are used as the input variables (commonly called "features") for each of the trained ANN models.

MLS brightness temperatures provide the means for the profile retrievals of various atmospheric properties and trace gas concentrations. Here, retrieved profiles of temperature ($T$), as well as concentrations of $H_2O$, $O_3$, $CO$, $SO_2$, nitric acid ($HNO_3$), and nitrous oxide ($N_2O$) provide the output variables (commonly called "labels") for each ANN model. The MLS level 2 (L2) Geophysical Product files report the respective operational profile retrievals; we use the most recent data, version 5 (Livesey et al., 2022). The spatial resolution of the L2 products depends on the species of interest, but typical values are $3\,\mathrm{km}$ in the

vertical and 5 and $500\,\mathrm{km}$ in the cross-track and along-track dimensions, respectively. The along-track distance between adjacent profiles is $\approx 165\,\mathrm{km}$. Only valid data, following the detailed data screening rules provided in Livesey et al. (2022), are considered. Information on the species-specific time range considered for training the ANN, as well as the employed MLS bands, channels, and MIFs used as input for the ANNs, are summarized in Table 1.

    Results of the ANN algorithm are also compared to those of the previous NRT retrievals based on optimal estimation (OE-

NRT). The OE-NRT retrievals are based on a modified L2 algorithm, which is necessary to reduce the data and computational resources. This imposes a number of limitations on the NRT products, such as a reduced number of valid profile retrievals and limitations on the recommended pressure ranges. Individual screening rules and recommendations are provided in Lambert et al. (2022); note that since January 2023 all MLS NRT data products are based on this new approach (ANN-NRT).

## 3   Artificial neural network

This section described the theory, training process, settings, performance evaluation, and data quality assessment of the updated, ANN-based NRT algorithm. The goal is to train ANN models on all valid MLS L2 standard product retrievals over 01/01/2005–04/30/2022 and their associated, nearest brightness temperature profiles. Since the MLS L2 standard products are used as labels (i.e., "truth") during training, the best-case output of each ANN is a computationally-inexpensive, high-fidelity preview of the L2 profiles.

### 3.1   Theory and general setup

A feedforward ANN is a type of machine learning model that consists of sequential layers that contain a large number of connected neurons, where the information only gets propagated forward from layer to layer. Propagating information backwards is not permitted. A more in-depth description of ANN setups and the involved mathematics can be found in, e.g., Reed and Marks (1999), Goodfellow et al. (2016), and Werner et al. (2021). Similar to the latter study, the model setup and determination of

model weights are facilitated by the "Keras" library for Python (version 2.2.4; Chollet et al., 2015), with "TensorFlow" (version 1.13.1) as the backend (Abadi et al., 2016).

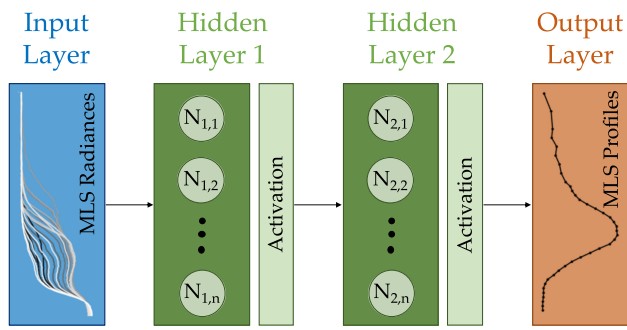

**Figure 1.** Simplified sketch of the algorithm setup.

A simplified sketch of the general model setup is shown in Figure 1. Note that the actual setup for each individual ANN-NRT model is notably more complex. The input layer, shown in blue, contains an $m \times n$ matrix of $n$ features sampled at $m$ different times and/or locations. In this study, the features are $n$ MLS brightness temperatures from individual spectral bands, channels, and MIFs from $m$ different MAFs (see Table 1 for the model-specific details). An example of a single MAF of MLS band 2 radiances is illustrated in Figure 1; the transition from black to white colors indicates the profiles sampled in channels 1–25. Each feature in the input layer is connected to individual neurons in the first hidden layer ($N_{1,j}$, $j = [1, 2, \cdots, J]$), shown in green. Each neuron value is derived as a linear superposition of the weighted input features. A subsequent activation layer introduces a degree of non-linearity. The simplified model in Figure 1 consists of a second hidden layer that contains neurons $N_{2,j}$, $n = [1, 2, \cdots, J]$. Here, each neuron value is calculated as a linear superposition of the weighted neuron output of the first hidden layer, after it passed through the first activation layer. Finally, following a second activation layer, there is the output layer (shown in dark orange), which consists of an $m \times k$ matrix of $k$ different labels. Here, the labels are values from individual profiles of a specific MLS retrieved L2 atmospheric constituent. Therefore, the size of $k$ is determined by the number of retrieval levels of the respective MLS L2 product. An example of a single $O_3$ profile is shown in Figure 1. As before, each neuron $N_{2,j}$ in the second hidden layer is connected to each of the $k$ labels by means of individual weights.

A detailed description of the training procedure is given in Werner et al. (2021). The necessary steps include randomly splitting the complete data set into training, validation, and test data ($75, 20, 5\%$ for each model in this study), determining the optimal hyperparameters via $k$-fold cross-validation, and the final training and validation of the model with the best set of hyperparameters. The hyperparameters that were considered in each model setup, some of which are described in more detail below, are (i) the number of hidden layers ($J_{HL}$), (ii) the number of neurons per hidden layer ($J_N$), (iii) the activation function (AF) employed in the activation layer, (iv) the amount of regularization, either via weight decay (i.e., the L2 regularization parameter; LRP) or alternatively the standard deviation of an extra Gaussian noise layer (GNS), and (v) the mini-batch size (MBS). The variables $n_{HL}$ and $n_N$ determine the complexity of the model. The choice of AF specifies the non-linear mathematical transformation of the individual neuron output. Introducing an LRP is one method to introduce regularization during the ANN training, which usually improves generalization of the model predictions for previously unseen data. Another method is to add Gaussian noise to each neuron input; the standard deviation of the noise added directly impacts the level of regular-

ization. During the training process the model weights are determined by iteratively minimizing a predefined loss function (the root-mean-square error, RMSE, in this study). Instead of using the full training data set during each iteration, only a random subset of the training data is used, determined by the parameter MBS. This approach not only improves generalization of the models (due to the introduced noise when minimizing the loss function), but also speeds up the training process.

Three additional hyperparameters that are not listed here are the choice of optimizer that minimizes the loss function during training; the learning rate, which affects the speed of convergence during training; and the number of "epochs", which is the number of iterations during training. We found that "Adam" optimization with a learning rate of $10^{-5}$ yielded the best model performance for each of the NRT species. Each model was trained with $\approx 10,000$ epochs, and the lowest validation loss was recorded. The ideal model weights are those associated with the minimum validation loss. Additional information about hyperparameters and their impact on model performance is given in, e.g., Reed and Marks (1999) and Goodfellow et al. (2016).

We considered the following ranges and settings: $J_{\mathrm{HL}} = [1, 2]$, $J_{\mathrm{N}} = [100, 200, \cdots, 2/3 \cdot (n+k)]$ per hidden layer, AF=["relu", "tanh"], LRP=[n/a, 1e$^{-6}$, 5e$^{-6}$, 1e$^{-5}$, $\cdots$, 1e$^{-1}$], GNS=[n/a, 1e$^{-3}$, 5e$^{-3}$, 1e$^{-2}$, $\cdots$, 1], and MBS=[32, 64, $\cdots$, 8192].

The computational costs associated with the training procedure of each ANN-NRT model, while dependent on the respective hyperparameters and size of the $m \times n$ input matrix, are generally as follows: it takes about one month to develop and train each ANN, using 12 CPUs and requiring $\approx 100$ GB of memory.

## 3.2 Hyperparameters and performance metrics for each model

Table 1 gives an overview of the ideal hyperparameters for each NRT species, determined after a comprehensive training procedure. It also provides details on the features that make up the input matrix for each ANN-NRT model, namely the start and end dates that define the training data record for each model, the number of total samples in that data record (determined by the number of successful profile retrievals), and the respective MLS bands, channels, and MIFs. Note that the MIFs for all models basically cover the vertical range of $\approx 400 - 0.001$ hPa. Since the models for each of the target species were developed separately, the end dates for the employed training data vary slightly. The choice of bands and channels was based on the absorption characteristics of each target molecule, as well as possible interference of other species.

Note that the model setups for $T$, CO, and SO$_2$ differ from those of the other species. The $T$ model is considerably more complex with comparatively high values of $J_{\mathrm{N}} = 5,078$ and MBS=8,192. The ANN-based estimator for temperature was developed before those for the other products, with less regard for computational cost than was present in the subsequent development. The computationally more expensive temperature model is "overbuilt", but had already been trained so was used in this version of the NRT products.

MLS mid-stratospheric observations of CO are basically just noise, which negatively affected model performance in the upper troposphere/lower stratosphere (UTLS) and in the upper stratosphere/mesosphere, where CO signals are stronger. The CO NRT product is of particular interest in the UTLS. As a result, we decided to train two different CO models: one for the four MLS retrieval levels in the UTLS between 215 and 68 hPa, and a second one for all other levels (including noisy levels in the middle stratosphere). The final CO profile predictions are a combination of both models.

**Table 1.** Summary of input features and hyperparameters for each ANN model. See text for more details.

| | Data Record | Samples | Bands | Channels | MIFs | $J_{HL}$ | $J_N$ | AF | LRP | GNS | MBS |
|---|---|---|---|---|---|---|---|---|---|---|---|
| $T$ | 01/01/2005–05/31/2021 | 19,084,479 | 1<br>8<br>22 | 1–25<br>1–25<br>40–90 | 23–97 | 2 | 5,078 | relu | n/a | $10^{-1}$ | 8,192 |
| $H_2O$ | 01/01/2005–03/31/2022 | 20,830,018 | 1<br>2<br>3<br>23 | 1–25<br>1–25<br>1–25<br>40–90 | 23–126 | 2 | 400 | tanh | $5^{-4}$ | n/a | 32 |
| $O_3$ | 01/01/2005–04/30/2022 | 21,296,092 | 1<br>7<br>8<br>24 | 1–25<br>1–25<br>1–25<br>40–90 | 23–126 | 2 | 400 | tanh | n/a | $10^{-1}$ | 32 |
| CO-UTLS | 01/01/2005–04/30/2022 | 15,959,662 | 8<br>9 | 1–25<br>1–22 | 23–56 | 2 | 1,068 | tanh | n/a | n/a | 32 |
| CO | 01/01/2005–04/30/2022 | 15,957,189 | 1<br>8<br>9<br>25 | 1–25<br>1–25<br>1–22<br>40–90 | 23–126 | 2 | 800 | tanh | n/a | n/a | 32 |
| $SO_2$ | 08/07/2008–09/02/2008<br>04/23/2015–05/07/2015<br>06/14/2009–07/19/2009<br>06/13/2011–06/28/2011<br>06/22/2019–08/18/2019<br>01/14/2022–01/28/2022 | 374,088 | 8 | 1–25 | 23–126 | 2 | 1,739 | relu | n/a | $10^{-1}$ | 32 |
| $HNO_3$ | 01/01/2005–08/31/2022 | 21,347,932 | 1<br>4<br>6<br>33 | 1–25<br>1–25<br>1–25<br>1–4 | 23–126 | 2 | 400 | tanh | $5^{-4}$ | n/a | 32 |
| $N_2O$ | 01/01/2005–08/31/2022 | 18,723,676 | 1<br>3<br>8 | 1–25 | 23–126 | 2 | 400 | tanh | $5^{-4}$ | n/a | 32 |

Similarly, MLS $SO_2$ retrievals at all stratospheric levels can be considered noise under standard atmospheric conditions. Elevated values are observed in air masses perturbed by volcanic eruptions. As a result, the $SO_2$ model was developed with a reduced data set covering periods of volcanic activity, namely the eruptions of Kasatochi, Calbuco, Sarychev, Nabro, Raikoke, and Hunga Tonga-Hunga Ha'apai (e.g., Pumphrey et al., 2015; Millán et al., 2022). An explanation to justify this decision is given below.

The hyperparameters reported in Table 1 are the ANN-NRT settings associated with the models that exhibited the highest performance scores during the training process. These scores were derived by comparing the ANN-NRT predictions with the respective MLS L2 results for all MAFs in both the validation and an independent test data set. The distinction between the two is important. Following the discussion in Ripley (1996) and Russel and Norvig (2009), the validation data is used for hyperparameter tuning and to prevent overfitting during model training. To truly evaluate the performance of a trained
model, a completely independent test data set is necessary. However, the performance scores for the validation and test data set should be similar and large discrepancies are an indication that the trained model does not generalize well (i.e., the model performs worse for previously unseen data). Note that of the $\approx 3500$ daily profiles MLS observed since 01/01/2005, $\approx 875$ and $\approx 175$ randomly selected samples are included in the validation and test data set, respectively. This means that Three specific scores were considered: Pearson's product-moment correlation coefficient ($R$), the root-mean-square deviation (RMSD), and
the median of the relative deviation between the derived ANN-NRT prediction and the L2 product (i.e., the bias).

     The performance metrics derived for the validation and independent test data set for each of the different ANN-NRT models are presented in Table 2. Since each of the MLS constituents describes a profile retrieval, the average over all valid retrieval levels is reported. With the exception of the $SO_2$ predictions, the average $R$ and absolute biases for the test data set are $> 0.72$ and $< 0.66\%$, respectively. The ANN models designed to predict $T$, $H_2O$, and $O_3$ perform particularly well, with $R > 0.88$,
RMSD$< 13\%$, and biases$< 0.32\%$. The very close agreements between the individual validation and test scores demonstrate that the derived models generalize well. As mentioned in section 2, stratospheric L2 retrievals in the absence of elevated levels of $SO_2$ can be considered noise, and comparisons between L2 and ANN-NRT results are difficult ($R = 0.26$ and bias $> 11\%$). If the training data set is increased to include all MLS retrievals between 01/01/2005 and 04/30/2022 (named $2^{nd}$ model in Table 2) rather than being restricted to volcanic activity, the associated correlation coefficients and biases slightly improve
to $0.37$ and $< 7\%$, indicating a better ability to predict noise. However, further analysis indicates that this model performs slightly worse for profiles containing elevated $SO_2$ concentrations; correlation coefficients for such profiles in the test data set are decreased by about 0.05 ($R = 0.52$ compared to $R = 0.57$), while the RMSD increases by about 0.31 ppbv (5.72 ppbv compared to 5.41 ppbv). Since the main objective of the $SO_2$ NRT is to detect volcanic activity, we decided to employ the model trained on the reduced (volcanic only) data set.

**3.3    Data quality assessment**

The OE-NRT retrieval provides numerous diagnostic quantities, similar to the operational MLS retrieval algorithm (Livesey et al., 2006), such as the estimated precision, status, and convergence, as well as an overall quality flag. Unfortunately, none of these quantities are available from the ANN predictions. Indeed, standard implementations of feedforward ANNs do not

**Table 2.** Summary of performance metrics for the validation data set, as well as an independent test data set for each of the ANN-NRT models, namely the average correlation coefficient ($R$), the average root-mean-square deviation (RMSD), and the average bias. Averages are calculated over all valid pressure levels. Percentages for both the RMSD and bias are calculated by normalizing by the average L2 value at each level.

| | Validation Data | | | Test Data | | |
|---|---|---|---|---|---|---|
| | $R$ | RMSD | Bias | $R$ | RMSD | Bias |
| $T$ | 0.96 | 1.65 K (0.77%) | 0.01 K ($< 0.01\%$) | 0.96 | 1.66 K (0.77%) | 0.01 K ($< 0.01\%$) |
| $H_2O$ | 0.87 | 7.75 ppmv (13.02%) | 0.32 ppmv (0.31%) | 0.87 | 7.52 ppmv (12.66%) | 0.32 ppmv (0.32%) |
| $O_3$ | 0.95 | 0.12 ppmv (9.85%) | $< 0.01$ ppmv (0.06%) | 0.95 | 0.12 ppmv (9.86%) | $< 0.01$ ppmv (0.06%) |
| CO-UTLS | 0.72 | 0.14 ppbv (24.43%) | $< 0.01$ ppbv (0.16%) | 0.72 | 0.14 ppbv (24.43%) | $< 0.01$ ppbv (0.16%) |
| CO | 0.74 | 0.40 ppbv (69.53%) | $< 0.01$ ppbv (0.49%) | 0.74 | 0.40 ppbv (69.42%) | $< 0.01$ ppbv (0.48%) |
| $SO_2$ | 0.27 | 5.52 ppbv (111.81%) | -0.05 ppbv ($-56.76\%$) | 0.26 | 5.45 ppbv ($-206.99\%$) | 0.05 ppbv ($-11.89\%$) |
| $SO_2$ (2$^{nd}$ model) | 0.37 | 4.88 ppbv ($-1065.92\%$) | $< 0.01$ ppbv (6.17%) | 0.37 | 4.87 ppbv ($-419.34\%$) | $< 0.01$ ppbv (5.73%) |
| $HNO_3$ | 0.75 | 0.56 ppbv ($-101.65\%$) | $< 0.01$ ppbv (0.83%) | 0.75 | 0.56 ($-6.42\%$) | $< 0.01$ ppbv ($-0.66\%$) |
| $N_2O$ | 0.89 | 7.95 ppbv (91.77%) | -0.02 ppbv (0.07%) | 0.89 | 7.95 ppbv (93.03%) | -0.02 ppbv (0.01%) |

provide any metrics for uncertainty quantification. ANN uncertainty comprises epistemic uncertainty, associated with limitations in the data set (i.e., not enough years to represent all possible atmospheric states), and aleatoric uncertainty, associated with uncertainties in the features and labels the model was trained on (i.e., measurement uncertainties in the MLS-observed brightness temperatures and retrieval uncertainties in composition profiles). Note that the retrieval uncertainties for the labels comprise uncertainties in the forward model and the prior assumptions.

Uncertainties in the ANN-NRT predictions for each composition profile value are derived by calculating the root sum square of (i) the typical MLS L2 precisions for the given pressure level taken from the training data set, and (ii) the RMSD between

the MLS L2 products and the predictions for the independent test data set. Negative precisions are assigned to values outside the valid pressure range, profiles in overlap regions (see Lambert et al., 2022), as well as those containing invalid radiances. Data values with negative precisions should not be used.

An additional data quality check assures that predictions at each pressure level are within a predefined confidence range. This range is derived from the minimum and maximum of the MLS L2 composition retrievals at each retrieval level, taken from the combined training, validation, and test data set. If a profile contains a prediction, at any level, that is smaller (bigger) than the minimum (maximum) value all the associated precisions are set to be negative. In other words, extrapolations by the ANNs are not permitted. Other MLS data quality metrics like status, convergence, and quality are not used.

## 4 Results

This section presents comparisons between MLS L2 profile retrievals and the respective OE-NRT and ANN-NRT predictions. These observations were made after the respective ANN-models were developed, trained, and evaluated and serve as examples of model performance going forward.

### 4.1 Statistical comparison with MLS L2

Figure 2a and Figure 2c show joint histograms of the OE-NRT and L2 $T$ retrievals at $21.54\,\text{hPa}$ (in the middle stratosphere) and $100.00\,\text{hPa}$ (in the UTLS). Data are from MLS observations over 1–31 July 2021, a period not employed in the ANN-NRT training process. Similar comparisons between the ANN-NRT predictions and L2 retrievals are shown in Figure 2b and d. Not only are the ANN-NRT distributions narrower at both of the levels shown, but also there are fewer outliers far away from the 1:1 line. Compared to the OE-NRT results, the ANN-NRT predictions exhibit higher correlation coefficients ($R = 0.98, 0.99$ vs. $R = 0.99, 1.00$ for 100.00 and $21.54\,\text{hPa}$, respectively) and a smaller range of minimum/maximum deviations from the L2 results.

Similar joint histograms for $H_2O$ are shown in Figure 2e-h. Because this ANN-NRT model was trained well after the $T$ model and the training data includes MLS observations sampled as late as April 2022, the comparisons shown here are for 1–31 May 2022. This provides the means to (i) assess ANN-NRT performance for previously unseen data and (ii) evaluate the ability of ANN-NRT to reproduce the unprecedented $H_2O$ enhancements in the persistent Hunga Tonga-Hunga Ha'apai plume (e.g., Millán et al., 2022). The $H_2O$ distribution at $21.54\,\text{hPa}$ reveals a significant underestimation in the OE-NRT retrievals for profiles with $H_2O > 8\,\text{ppmv}$ associated with the volcanic plume. In contrast, the ANN-NRT can reliably predict values of up to 16 ppmv. At $100.00\,\text{hPa}$, the ANN-NRT distribution is noticeably narrower, with fewer outliers off the 1:1 line compared to the OE-NRT results. At the $100\,\text{hPa}$ pressure level, the ANN-NRT predictions have a significantly higher correlation coefficient than the OE-NRT retrievals ($R = 0.80$ compared to $R = 0.66$), while the 1[st] and 99[th] percentile of the differences with L2 are reduced (0.9 ppmv compared to 1.3 ppmv). At the $21.54\,\text{hPa}$ level both NRT products exhibit $R = 0.98$.

Comparisons of L2, OE-NRT, and ANN-NRT $O_3$ are shown in panels (i)-(l). The OE-NRT algorithm performs well at both levels, with $R = 1.00$ and only a few obvious outliers observed, while ANN-NRT provides similarly good performance

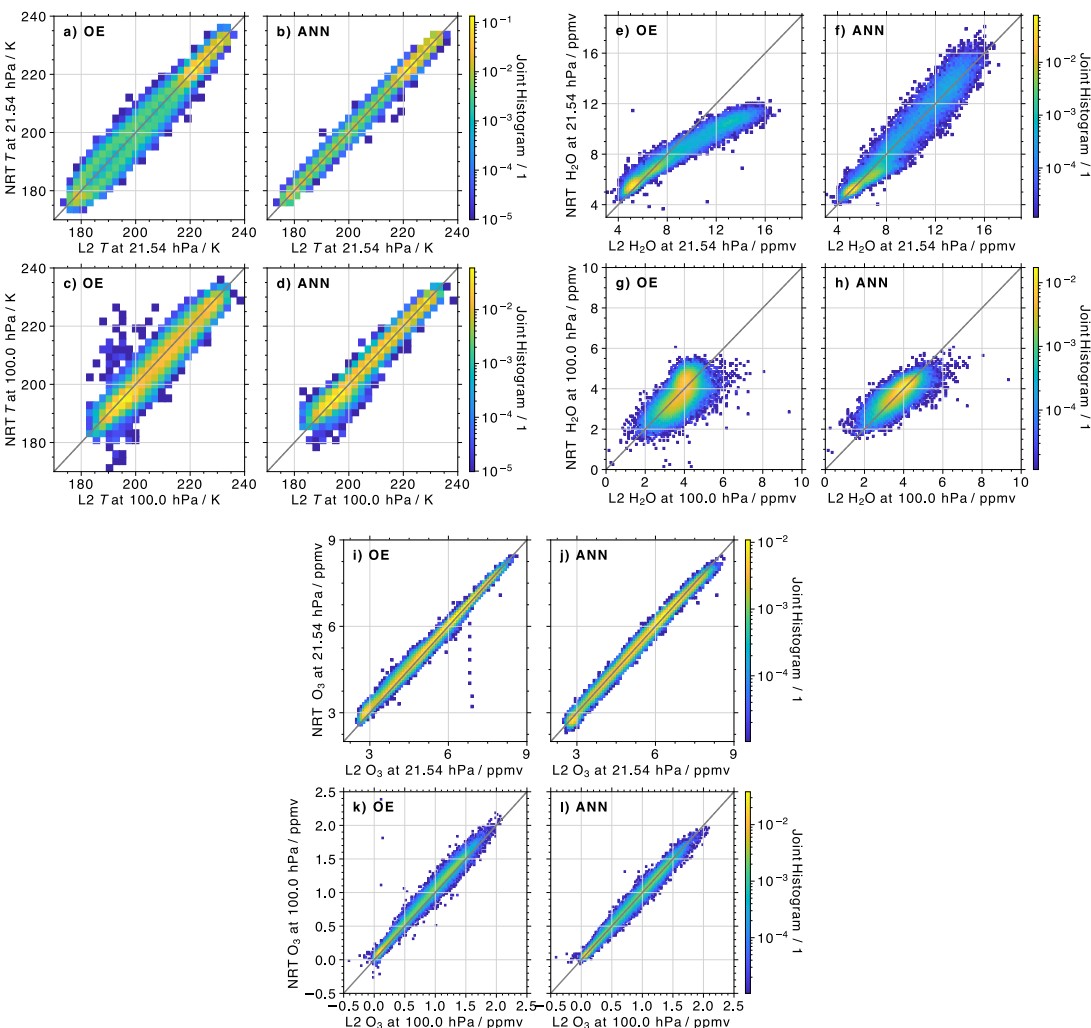

**Figure 2.** (a) Joint histograms of $T$ derived from OE-NRT and L2 at 21.54 hPa. Data are from MLS observations over 1–31 July 2021. The gray diagonal line indicates 1:1 correlation. (b) Similar to (a), but showing joint histograms of the ANN-NRT and L2 results. (c)-(d) Same as (a)-(b), but at 100.00 hPa. (e)-(h) and (i)-(l) Similar to (a)-(d), but for $H_2O$ and $O_3$ over 1–31 May 2022.

($R = 1.00$ at both levels). Joint histograms between L2 retrievals and the OE-NRT results, as well as the ANN-NRT predictions for CO, $SO_2$, $HNO_3$, and $N_2O$, are shown in Figure A1 in the appendix.

220    Figure 3 presents profiles of three metrics that characterize the performance of the two NRT algorithms. Panels (a)-(c) show derived $R$, RMSD, and bias between $T$ from L2 and OE-NRT (red), as well as between L2 and ANN-NRT (blue). At all retrieval levels, the ANN-based $T$ predictions have higher $R$ ($> 0.950$) and lower RMSD ($< 3.4\%$). The ANN-NRT bias shows little vertical variability and is within $\pm0.3\%$ at all levels, whereas the OE-NRT bias shows some oscillatory behavior and much larger variability (values within $\pm1.5\%$).

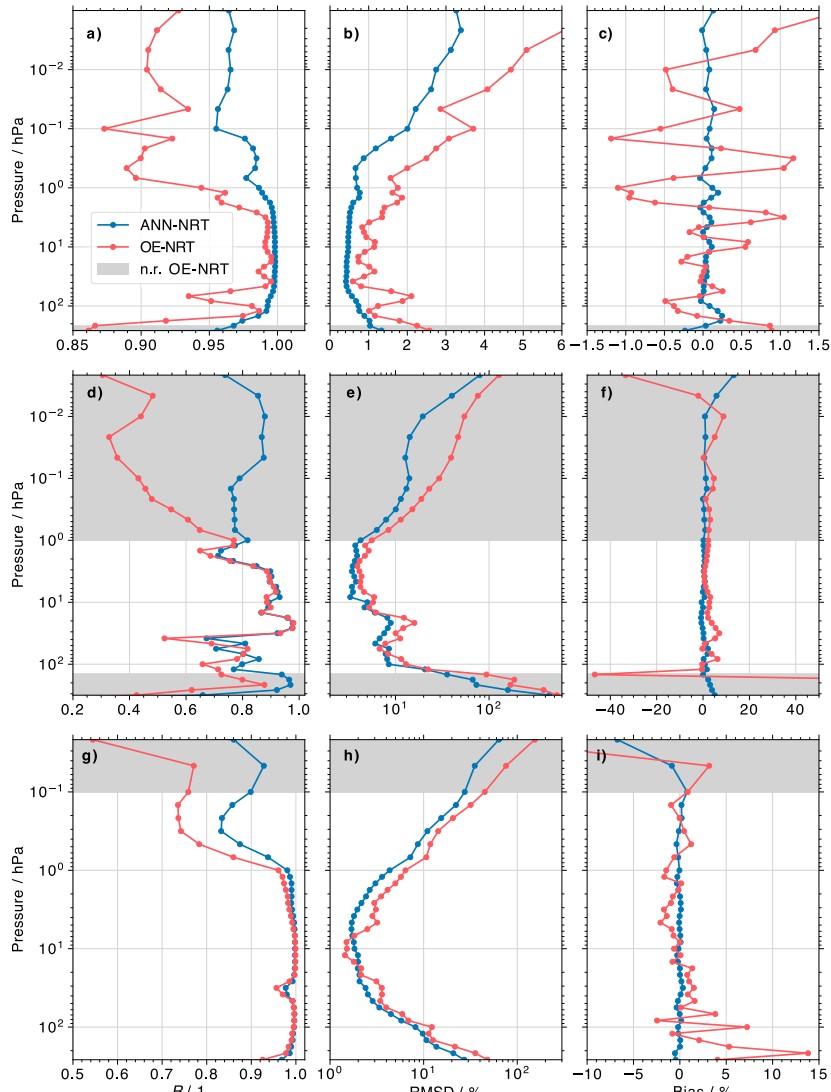

**Figure 3.** (a) Profiles of correlation coefficient ($R$) between OE-NRT and L2 $T$ (red), as well as the ANN-NRT and L2 results (blue). Data are from MLS observations over 1–31 July 2021. The vertical extent is defined by the recommended L2 data screening procedures; gray areas indicate levels at which the OE-NRT product is not recommended for scientific use. (b)-(c) Same as (a), but showing the root-mean-square deviation (RMSD) and bias, respectively. Both the RMSD and bias are normalized by the average L2 $T$ at each level. (d)-(f) and (g)-(i) Similar to (a)-(c), but for $H_2O$ and $O_3$, respectively, over 1–31 May 2022.

225    The recommended range for the OE-NRT $H_2O$ retrievals is $147$–$1$ hPa. Here, the performance metrics for the ANN-NRT predictions compare well to those of the OE-NRT retrievals, and the derived $R$, RMSD, and bias values are very similar (panels (d)-(f)). Outside of that range the OE-NRT performance degrades noticeably and ANN-NRT yields more reliable $H_2O$

values that are closer to the L2 retrievals. Here $R$ is $> 0.75$, RMSD is $< 65\%$, and the bias is within $15\%$. In the case of the O$_3$ retrievals (panels (g)-(j)), the derived $R$ values for the OE-NRT and ANN-NRT algorithms are very similar. Only above $\approx 1\,\mathrm{hPa}$ does the OE-NRT performance suffer, and the correlations between the L2 and the ANN-NRT results are more than 0.1 higher. At almost all retrieval levels, the ANN-NRT exhibits slightly smaller RMSD and biases compared to the OE-NRT algorithm. Similar profiles for CO, SO$_2$, HNO$_3$, and N$_2$O are shown in Figure A2 in the appendix.

A summary of average performance metrics is given in Table 3, derived for the same time period as is used in Figs. 2–3 and Figs. A1–A2. Specifically, the presented metrics are: $R$, the average absolute RMSD, and average absolute bias for each species and the two NRT algorithms, as well as the averages of the $1^{\mathrm{st}}$ and $99^{\mathrm{th}}$ percentile of the differences to L2 (as a proxy for the minimum and maximum deviations). Averages are calculated over all valid pressure ranges (excluding levels not recommended for OE-NRT). Note that two sets of SO$_2$ statistics are shown: one set based on MLS observations in January 2022, which are affected by the Hunga Tonga-Hunga Ha'apai volcanic eruption and were included in training data set, and a second set based on samples in May 2022 with no volcanic influence. Except for the stratospheric CO, N$_2$O, and HNO$_3$ models, the ANN-NRT predictions always exhibit higher $R$, lower RMSD, lower biases, and lower minimum and maximum differences to L2. These three species are sampled at a number of stratospheric levels where the retrieved concentrations are very close to zero and can be considered noise. As illustrated in Figures A1 and A2, the OE-NRT algorithm statistically fits that noise better than the ANN-NRT models. Apart from the noisy retrieval levels, the ANN-NRT approach provides profile predictions that agree better with the operational MLS L2 data products.

## 4.2   Global maps for individual example days

Figure 4a presents global maps of temperatures provided by the operational MLS L2 algorithm (left column), the OE-NRT product (middle column), and the ANN-NRT predictions (right column). Data are from 12 July 2021, a representative example day that was not part of the training data set and thus unseen by the ANN-NRT model. Each temperature product is shown at two different levels: at $100.00\,\mathrm{hPa}$ in the UTLS (bottom panels) and at $21.54\,\mathrm{hPa}$ in the middle stratosphere (top panels). At both levels the three data products provide similar results, and both the OE-NRT and ANN-NRT algorithm reproduce the general patterns observed in the L2 temperatures. Compared to the L2 results, the OE-NRT product exhibits an increased frequency of invalid retrievals, as reflected by the areas in white over the Southern Ocean.

Similar example maps for H$_2$O and O$_3$ on 22 May 2022 are shown in Figure 4b and 4c. At $100.00\,\mathrm{hPa}$ there are areas with strong overestimates of the H$_2$O from OE-NRT compared to L2 (dark blue colors), while concentrations in the tropics and subtropics are generally underestimated (light violet colors). Here, the ANN-NRT performs more reliably, and the results are closer to the L2 data. A notable exception is the area of increased H$_2$O over India and parts of Southeast Asia, where the ANN-NRT underestimates the L2-retrieved concentrations. This region is characterized by strong and deep convection during the monsoon months that affects the sampled radiance profiles and may introduce uncertainties into the ANN model predictions. Maps of $100.00\,\mathrm{hPa}$-H$_2$O concentrations on other days during that week indicate that slight underestimations persist in this area; however, the ANN-NRT predictions generally are much closer to the L2 results than are the OE-NRT retrievals. At the same $100.00\,\mathrm{hPa}$-level the OE-NRT algorithm also yields slight overestimates of tropical O$_3$, indicated by the

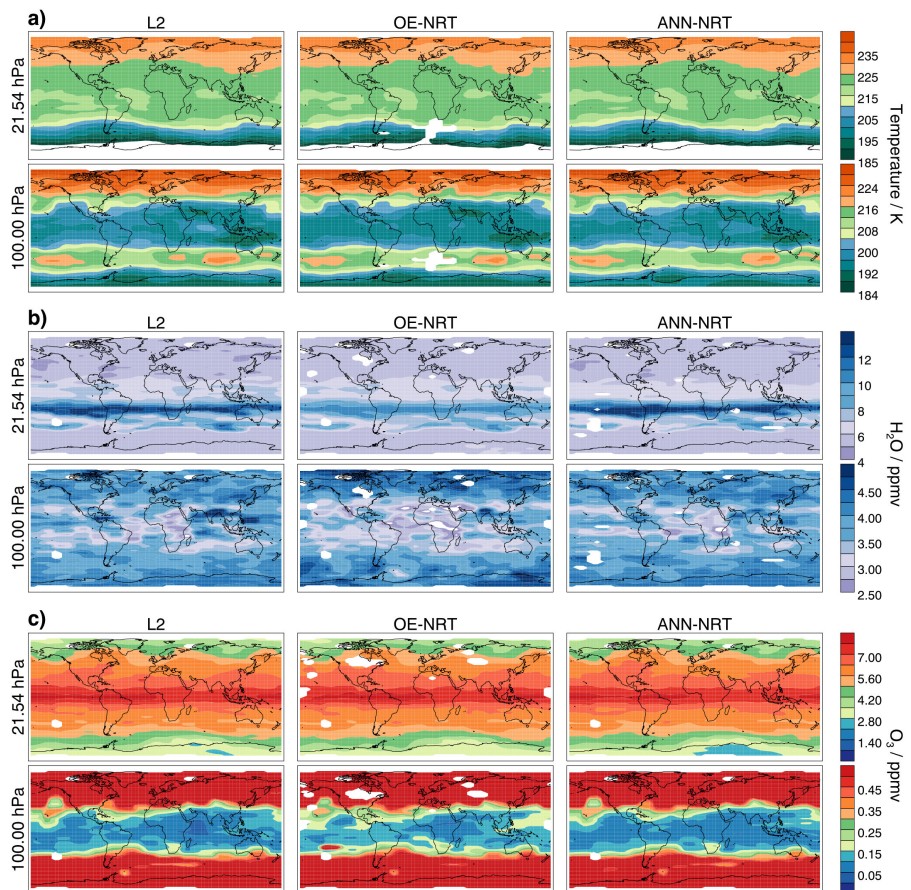

**Figure 4.** (a) Maps of derived $T$ provided by the MLS L2, OE-NRT, and ANN-NRT algorithm at two different levels on 12 July 2021. (b)-(c) Similar to (a), but for $H_2O$ and $O_3$, respectively, on 22 May 2022.

lighter blue colors. In the middle stratosphere at 21.54 hPa, the significant underestimates of tropical $H_2O$ from the OE-NRT retrievals is evident, which confirms the results seen in Figure 2e. The ANN-NRT algorithm is able to replicate the elevated L2 concentrations. At this level the $O_3$ concentrations from the two NRT approaches are very similar. The only obvious difference

is the area of low concentrations over Antarctica, which is completely missed by the OE-NRT algorithm and is overestimated (in area) by ANN-NRT. Note that profiles sampled in this region are affected by radiances that are reflected by the surface (see Fig. 7d in Werner et al., 2021 and the relevant discussion), which might impact the reliability of the ANN predictions. Similar maps for $CO$, $SO_2$, $HNO_3$, and $N_2O$ are shown in Figure B1 in the appendix.

## 5   ANN-NRT performance for different amounts of training data

The analysis in section 4 illustrates that the new ANN-NRT algorithm generally provides reliable results in closer agreement to the operational MLS L2 products (compared to OE-NRT). This shows that it is possible, potentially advisable, to employ machine learning techniques to obtain more reliable NRT data products for current and future mission concepts. However, the good performance of ANN-NRT may hinge on the long MLS data record, which encompasses more than 17 years of global observations. If ANN-based NRT approaches only provide reliable results when trained on extensive data sets that only

become available after many years of observations, then machine learning might be a less attractive solution after all. In order to test how the amount of available training data affects the reliability of the ANN-NRT predictions, we calculated performance metrics for two of the ANN-NRT models in this study when trained with differently sized training data sets. Note that the training data size refers to all data involved in the training and evaluation procedure and thus also includes the validation and test data set. For the analysis in this section, the size of the training data was first set to one year, and subsequently doubled to

two, four, and eight years. The performance metrics derived for each of these models were then compared to the ones for the fully trained ANN-NRT algorithm, i.e., using the data records indicated in Table 1. We focus on the models for $T$ and $O_3$, i.e., quantities for which the OE-NRT algorithms perform comparatively poorly and well, respectively.

Figure 5 shows the average $R$, RMSD, and bias between the operational MLS L2 retrievals and both the OE-NRT and ANN-NRT results for the two species. Similar to the analysis in Figures 2 and 3, the comparisons are based on observations over

1–31 July 2021 ($T$) and 1–31 May 2022 ($O_3$). Averages (red lines and blue dots for OE-NRT and ANN-NRT, respectively) and standard deviations (blue error bars; for clarity only shown for the ANN-NRT predictions) are calculated over all valid pressure levels following the data screening procedures for the OE-NRT products, thus ignoring levels in the extended ANN-NRT range indicated in section 4.1. It is obvious that for both species, average $R$ values increase monotonically with increasing training data size, while the average RMSD monotonically decreases. At the same time, the standard deviation for each metric

slightly decreases. A very small increase in the averaged absolute biases for the $T$ models is observed. However, these absolute biases are in the range of 0.11–0.16 K (0.05–0.06 K if both positive and negative biases are averaged) and can be considered negligible. Note that similar analysis for the $1^{st}$ and $99^{th}$ percentile of the difference between MLS L2 retrievals and each ANN-NRT model prediction shows a monotonically decreasing behavior with increasing training data size.

Surprisingly, even if just a single year of observations is available to train the ANN-NRT $T$ model, the derived performance

metrics show a significant improvement when compared with the OE-NRT results. Here, $R$ increases from 0.95 to 0.98, the RMSD is reduced from 2.00% to 1.17%, and the absolute bias is reduced from 0.50% to 0.06%. Even for $O_3$, where the current NRT algorithm performs rather well, the ANN model trained on one year of MLS observations yields noticeable improvements. While the correlation coefficients and RMSD are comparable (0.95 vs. 0.94 and 9.93% vs. 10.10%), the absolute bias is reduced from 1.79% to 0.37%.

These results illustrate that the simplified OE-NRT retrieval algorithm could have been replaced by machine learning approaches as early as one year after the beginning of the mission, which would have resulted in more reliable NRT data products.

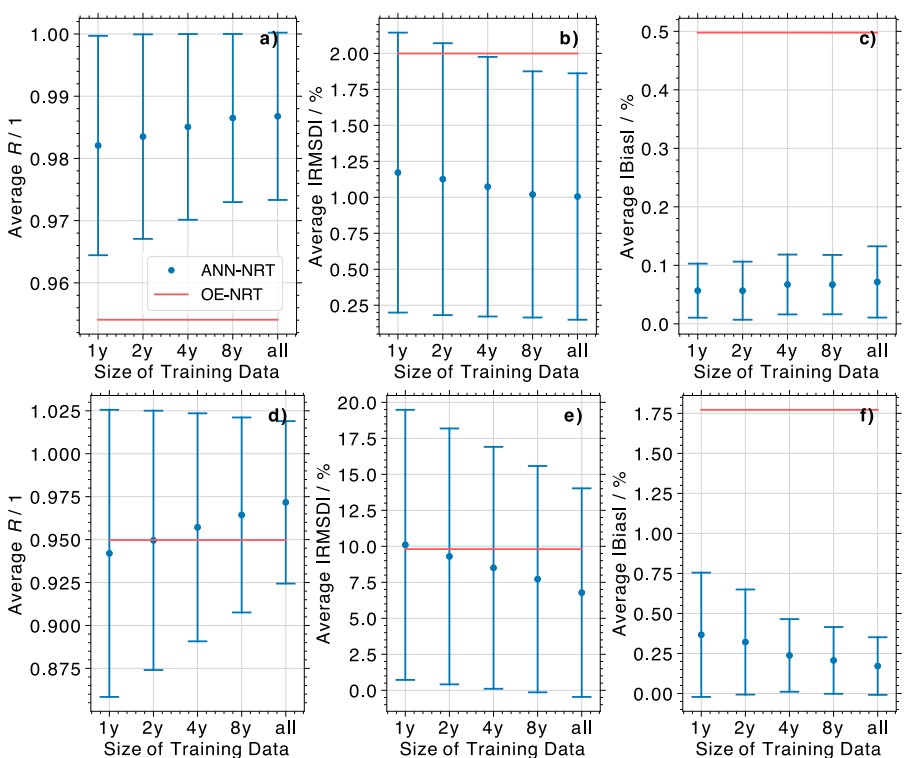

**Figure 5.** (a) Average correlation coefficient ($R$) between $T$ from the MLS L2 and OE-NRT retrieval algorithms (red line), as well as the L2 and ANN-NRT results (blue dots), for differently sized training data sets. Vertical bars indicate the range covered by $\pm 1$ standard deviation, based on the variability in $R$ for different retrieval levels. (b)-(c) Same as (a), but showing the average absolute root-mean-square deviation (RMSD) and bias. Both the RMSD and bias are normalized by the average L2 temperature at each level. (d)-(f) Similar to (a)-(c), but for ozone.

## 6  Conclusions

The previous version of MLS NRT data products (OE-NRT) is replaced with predictions from an artificial neural network (ANN). This manuscript describes the setup and evaluation of ANN models for all MLS NRT species. Starting in January 2023, all MLS NRT data products are based on this new approach (ANN-NRT).

The biggest improvements compared to OE-NRT are observed for $T$, water vapor ($H_2O$), and $O_3$. The analysis in this study shows that for these products the ANN-NRT algorithm yields noticeably higher correlation coefficients ($R$), as well as lower root-mean-square deviations (RMSD) and biases when compared to the operational L2 results.

The ANN-NRT predictions for carbon monoxide (CO), nitric acid ($HNO_3$), and nitrous oxide ($N_2O$) are characterized by good performance at most retrieval levels. However, the OE-NRT algorithm does a better job at fitting the L2 noise for concentrations close to $0\,\mathrm{ppbv}$. Here, ANN-NRT tends towards predicting $0\,\mathrm{ppbv}$ regardless of the L2 values, which might be the preferable behavior as it produces less noisy background concentrations.

Of special note is the ANN-NRT setup for sulphur dioxide ($SO_2$). Volcanic eruptions are the primary source of stratospheric $SO_2$. As a result, we decided to train the $SO_2$ ANN model on MLS observations around major volcanic eruptions, namely those of Kasatochi, Calbuco, Sarychev, Nabro, Raikoke, and Hunga Tonga-Hunga Ha'apai (e.g., Pumphrey et al., 2015; Millán et al., 2022). While ANN-NRT performs well in reproducing elevated $SO_2$ concentrations associated with the Hunga Tonga-Hunga Ha'apai eruption, the training data is limited and the model may suffer from overfitting (i.e., learning specific characteristics of known eruptions well to the detriment of generalization).

Global maps of predicted $H_2O$ and $O_3$ concentrations indicate that model performance may be affected by the presence of strong, deep convection, as well as by strong surface reflections over Antarctica. While the respective predictions agree better with the L2 retrievals compared to the OE-NRT results, more analysis is needed to explore potential improvements to the ANN setups.

Besides the better agreement with the operational L2 retrievals (compared to OE-NRT), the ANN-NRT approach is computationally more efficient. Current tests reveal that ANN-NRT provides data $\approx 5 - 12$ times faster than the OE-NRT algorithm.

The results presented in this manuscript indicate that, instead of relying on simplified retrieval algorithms and assumed approximations to provide timely NRT data products, machine learning approaches can be utilized to obtain results both more reliably and more rapidly. However, the application to MLS data benefits from the extended data record of more than 17 years of daily, global observations. A sensitivity study was performed to test the effects of significantly reduced amounts of training data on the reliability of predicted $T$ and $O_3$. ANN-NRT models were trained with 1, 2, 4, and 8 years of MLS observations, and the performance in each case was compared to results from the best models, which were trained on $> 17$ years of data. This simulates the process of training the ANN-NRT setup after 1, 2, 4, and 8 years of observations. It is shown that even models that were trained on only one year of MLS data outperform the OE-NRT algorithm, which demonstrates the potential of applying machine learning to generate NRT products for other current and future mission concepts with similar sampling frequency. Alternative approaches, like training ANNs on synthetic profiles of atmospheric constituents and simulated brightness temperatures, may be needed for instruments with significantly lower sampling rates.

*Data availability.* MLS L1 radiance data and L2GP data, including status flags, are available at https://disc.gsfc.nasa.gov. NRT data are available at https://www.earthdata.nasa.gov/learn/find-data/near-real-time/mls.

## Appendix A:  Statistical comparison with MLS L2: CO, $SO_2$, $HNO_3$, and $N_2O$

This section presents joint histograms (Figure A1) and profiles of performance metrics (Figure A2) derived for the CO, $SO_2$, $HNO_3$, and $N_2O$ retrievals from the three algorithms. These results complete the analysis described in section 4.1.

There are no CO sources in the middle stratosphere, and the MLS retrievals can be primarily considered noise. This is evident in Figure A1a, which shows a joint histogram of L2 and OE-NRT retrievals at 21.54 hPa. The distribution is centered around very low positive values, and almost all retrievals are in the range $-20$ to $40$ ppbv. A similar distribution of L2 and

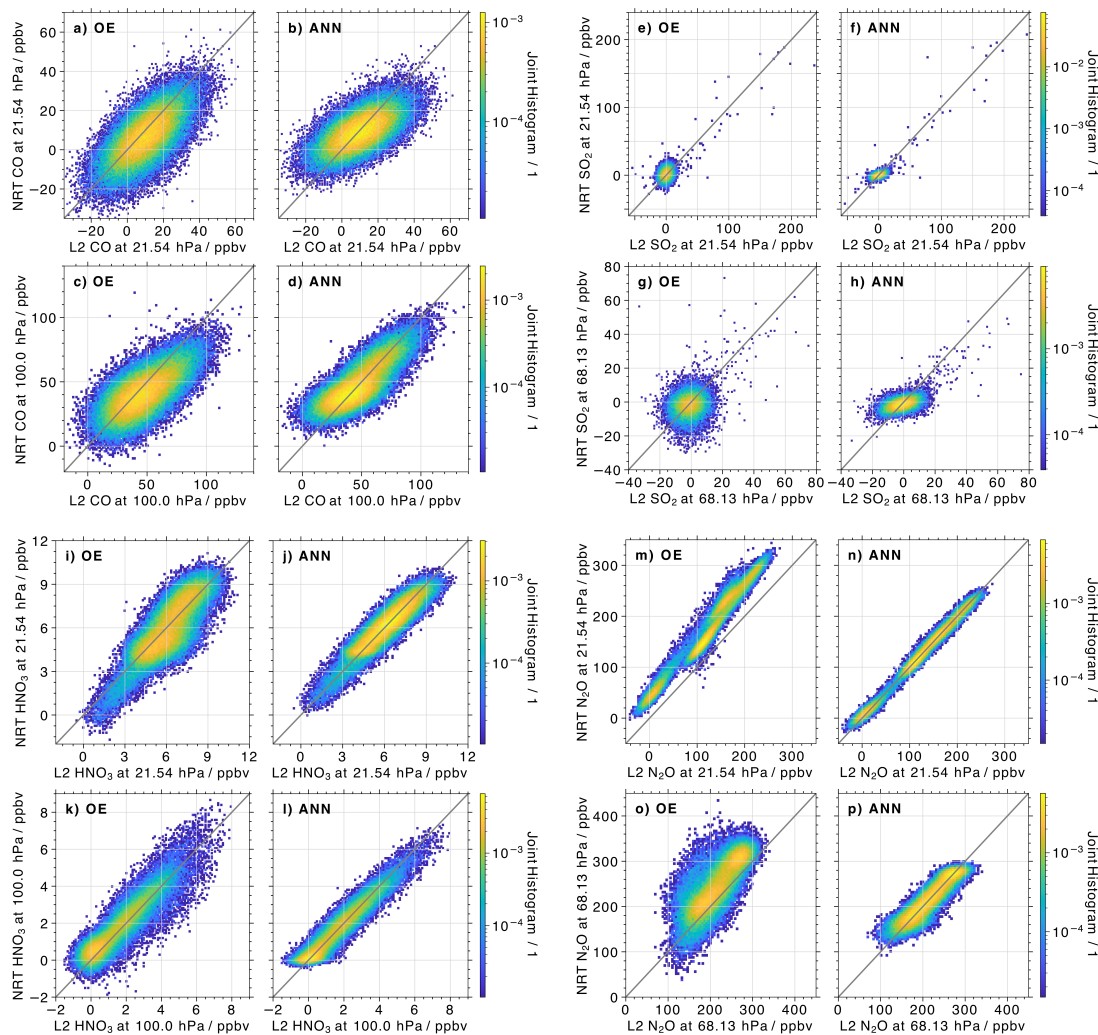

**Figure A1.** Similar to Fig.2, but for (a)-(d) CO over 1–31 May 2022, (e)-(h) SO$_2$ over 15–22 January 2022, as well as (i)-(l) HNO$_3$ and (m)-(p) N$_2$O over 1–30 September 2022.

ANN-NRT results is shown in panel (b), albeit with a slight tilt relative to the 1:1 line. The ANN-NRT $R = 0.51$ is slightly lower than the one for OE-NRT ($R = 0.55$). Noticeably higher CO concentrations are observed at 100.00 hPa; the respective joint histograms are shown in Figure A1c-d. Here, the ANN-NRT distribution shows values closer to the 1:1 line compared to the OE-NRT results, which indicates a higher correlation between the predictions and L2 retrievals ($R = 0.80$ vs. $R = 0.68$).

As mentioned in sections 2–4, background SO$_2$ concentrations in the stratosphere are essentially 0 ppbv and the MLS retrievals can be considered noise. However, air masses that are affected by volcanic eruptions show significantly enhanced concentrations. The joint histograms of L2 and OE-NRT, as well as L2 and ANN-NRT results, are shown in Figure A1e-h. Data are from 15–22 January 2022, the first week after the Hunga Tonga-Hunga Ha'apai eruption (e.g., Millán et al., 2022).

Each distribution is centered around concentrations of 0 ppbv, but individual MLS profiles show elevated concentrations of up to 200 ppbv (at 21.54 hPa) and 80 ppbv (at 68.13 hPa; this level was chosen to present profiles that are less affected by the volcanic eruption). The parts of the ANN-NRT distributions that resemble $SO_2$ noise are tighter and appear almost horizontal, indicating that the ANN-NRT tends to predict concentrations close to 0 ppmv independent of the L2 noise. Conversely, the distributions from the L2 and OE-NRT results appear random for the noisy part and slightly more scattered around the 1:1 line for observations in the volcanic plume. Correlation coefficients are higher for the ANN-NRT results, both in the middle stratosphere ($R = 0.86$ vs. $R = 0.70$) and in the UTLS ($R = 0.62$ vs. $R = 0.46$).

Figure A1i-l shows a clear improvement for the $HNO_3$ predictions based on the ANN-NRT model compared to the OE-NRT algorithm. The distributions are tighter, and fewer outliers are noticeable at both the 21.54 ($R = 0.92$ vs. $R = 0.83$) and 100.00 hPa ($R = 0.96$ vs. $R = 0.92$) levels. A similarly stark improvement from the ANN-NRT algorithm is evident for $N_2O$, indicated by the joint histograms in Figure A1m-p. Not only does ANN-NRT remove the noticeable bias that is evident in the OE-NRT results, but also the distributions are closer to the 1:1 line ($R = 0.99/R = 0.92$ vs. $R = 0.98/R = 0.81$ at 68.13/21.54 hPa). Note that MLS $N_2O$ retrievals are not recommended at 100.00 hPa.

Similar to earlier analysis, Figure A2 provides a more quantitative evaluation of the OE-NRT and ANN-NRT performance. Again, profiles of derived performance metrics from the MLS L2 products and the current OE- and ANN-based NRT results are presented.

While the ANN-NRT CO predictions exhibit slightly higher (lower) $R$ (RMSD) values in the UTLS and upper stratosphere, the ANN-NRT approach seems to do worse in the middle stratosphere between $\approx 46$ and $3.2$ hPa. At these levels, the CO retrievals can be considered noise, where the ANN-NRT tends to predict values closer to 0 ppbv regardless of the L2 value. Meanwhile, the ANN-NRT bias varies within 15% and shows fewer oscillations than the OE-NRT results.

The ANN-NRT performance metrics for $SO_2$ indicate a more reliable $SO_2$ prediction than from the OE-NRT algorithm, with better $R$, RMSD, and bias results at every retrieval level (note that the absolute values are plotted in panel e). This can be partly explained by the fact that 75% of MLS profiles sampled over 1–22 January 2022 are included in the training data set for the ANN-NRT model in order to focus on model reliability for air masses affected by volcanic eruptions. Predicting concentrations for observations over 1–31 May 2022 provides the means to evaluate ANN-NRT performance for previously unseen data, albeit for a time period without $SO_2$ enhancements due to volcanic influence. Compared to the OE-NRT results, the ANN-NRT predictions are characterized by higher $R$, as well as lower RMSD and biases, at all valid retrieval levels. As an example, the ANN-NRT algorithm (OE-NRT) exhibits $R = 0.34$ ($R = 0.22$) at 21.54 hPa and $R = 0.22$ ($R = 0.14$) at 68.13 hPa.

Apart from retrieval levels above $\approx 4.6$ hPa, the $HNO_3$ predictions from ANN-NRT compare better with the MLS L2 retrievals, indicated by higher $R$, as well as lower RMSD and bias values. This improvement is especially noticeable in the upper troposphere (pressures $> 100$ hPa), where the OE-NRT product is not recommended.

Similar to CO, there are pressure levels where the $N_2O$ retrievals can be considered noise (in the upper stratosphere for pressures below $\approx 5$ hPa). Here, the ANN-NRT results exhibit lower $R$ and higher RMSD. However, the bias remains small, with values within $\approx 10\%$.

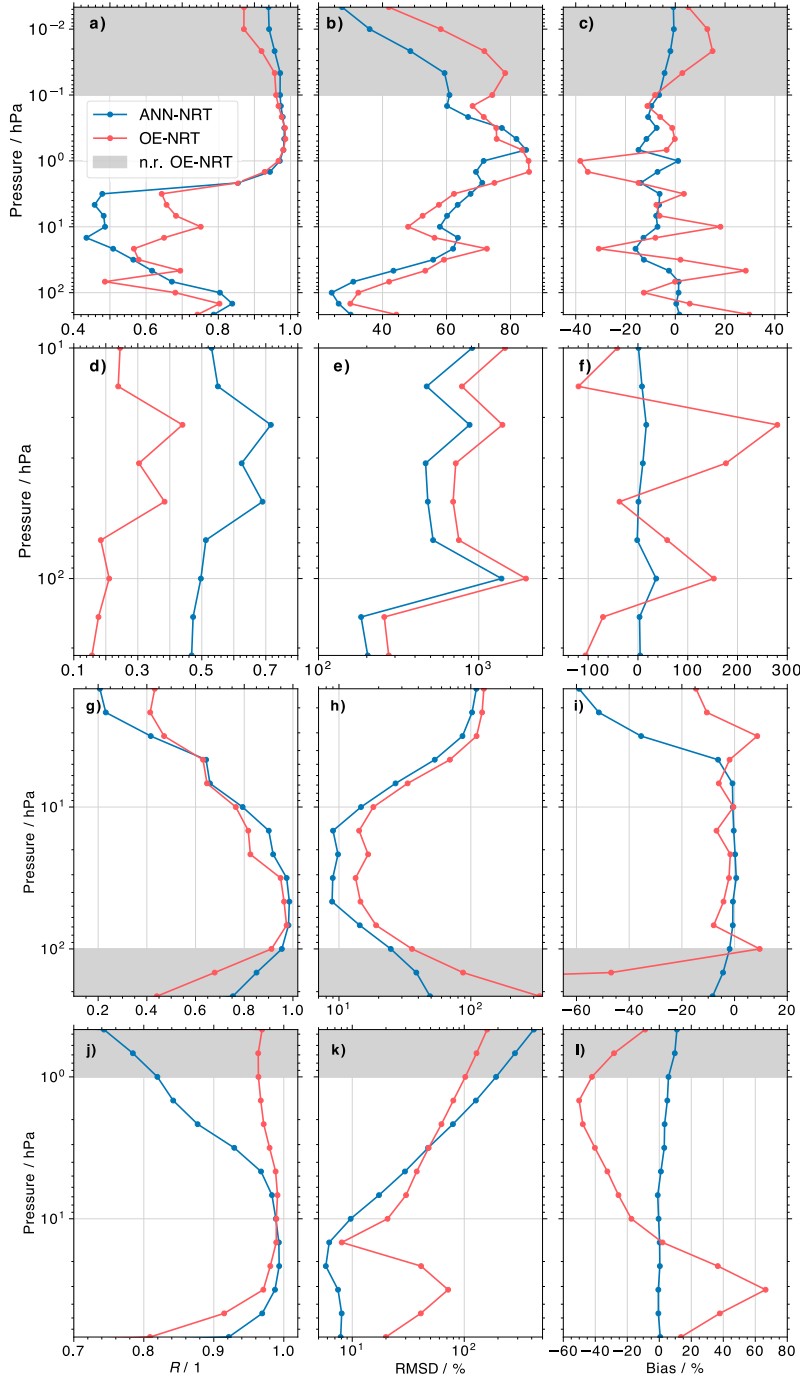

**Figure A2.** Similar to Fig.3, but showing performance metrics for (a)-(c) CO over 1–31 May 2022, (d)-(f) SO$_2$ over 15–22 January 2022, as well as (g)-(i) HNO$_3$ and (j)-(l) N$_2$O over 1–30 September 2022.

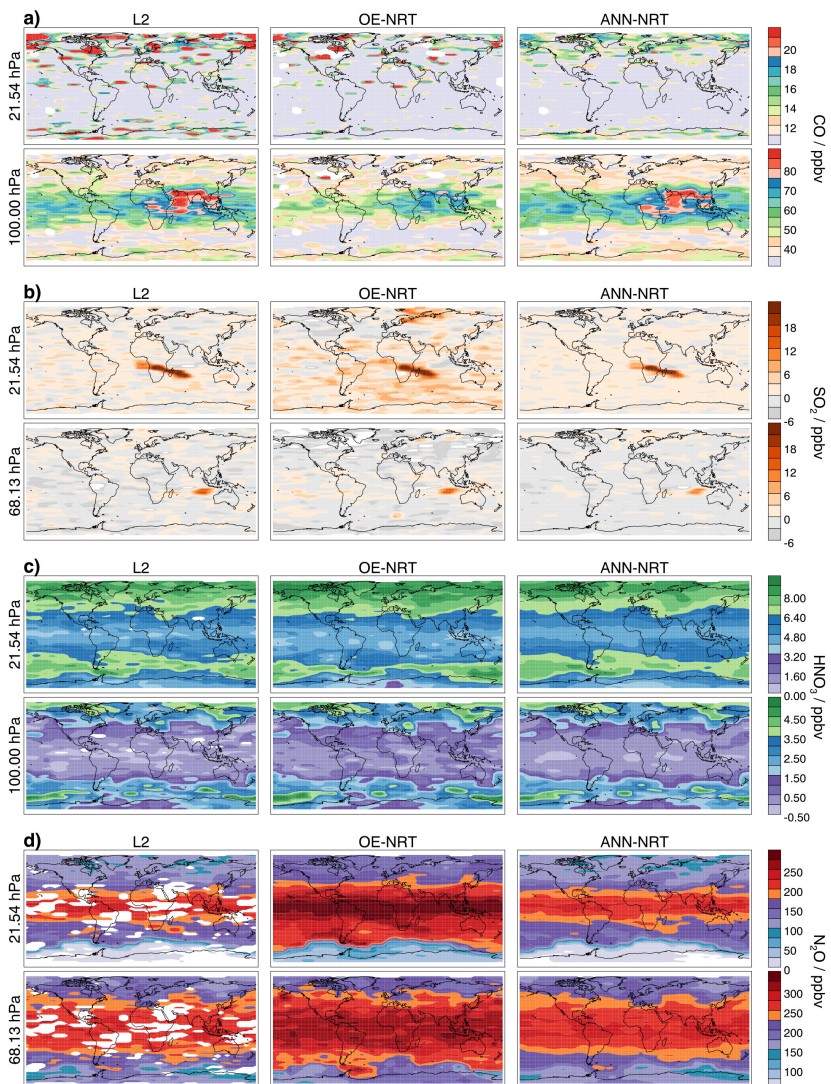

**Figure B1.** Similar to Fig.4, but showing maps of (a) CO on 22 May 2022, (b) $SO_2$ on 22 January 2022, as well as (c) $HNO_3$ and (d) $N_2O$ on 22 September 2022.

## Appendix B: Global maps for individual example days: CO, $SO_2$, $HNO_3$, and $N_2O$

This section presents global maps of CO, $SO_2$, $HNO_3$, and $N_2O$ from the three algorithms for representative example days (Figure B1) and completes the analysis in section 4.2.

Figure B1a shows CO on 22 May 2022 from the L2, OE-NRT, and ANN-NRT algorithms at $100.00$ hPa (bottom panels) and $21.54$ hPa. Two characteristics that were previously mentioned are noticeable: ANN-NRT outperforms the OE-NRT algorithm in the UTLS (see the enhanced concentrations in the region of the Asian summer monsoon; red colors), while it

predicts smoother $CO$ noise with concentrations closer to $0$ ppbv (see the absence of red colors in the Northern Hemisphere at $21.54$ hPa). Similar observations about the performance for noisy data can be made for the $SO_2$ example map, shown in

Figure B1b. At both retrieval levels, ANN-NRT reproduces the enhanced values over the Indian Ocean (at $68.13$ hPa) and over the African continent (at $21.54$ hPa), while predicted concentrations everywhere else are closer to $0$ ppbv (light gray and light salmon colors).

Differences between the OE-NRT and ANN-NRT algorithms are more subtle for the $HNO_3$ field, presented in Figure B1c. In the tropics and subtropics at $100.00$ hPa, the OE-NRT concentrations are slightly too low (compared to L2), as indicated

by the darker purple colors. Similar underestimations in the OE-NRT retrievals are noticeable at $21.54$ hPa, especially in the Southern Ocean west of South America and over Antarctica.

Significant differences are observed for the global $N_2O$ fields in Figure B1d. The OE-NRT retrievals exhibit strong overestimation (dark red colors) in the tropics, subtropics, and mid-latitudes. Likewise, concentrations in the polar regions are too high (dark purple colors). The ANN-NRT approach not only does a much better job at reproducing the L2 retrievals, but it also

does not suffer from the data gaps (white colors) apparent in the L2 data, which arise from the extensive screening rules..

**Table 3.** Summary of average correlation coefficient ($R$), average absolute root-mean-square deviation (RMSD), and average absolute bias, as well as the averages of the 1$^{st}$ and 99$^{th}$ percentile of the difference between the various OE-NRT and L2 products, and the ANN-NRT and L2 results. Percentages for the RMSD, bias, and percentile differences are calculated by normalizing by the average L2 value at each level. Averages are calculated over all all valid OE-NRT pressure levels.

| | $R$ | | RMSD | | Bias | | 1$^{st}$ perc. | | 99$^{th}$ perc. | |
|---|---|---|---|---|---|---|---|---|---|---|
| | OE | ANN | OE | ANN | OE | ANN | OE | ANN | OE | ANN |
| $T$ | 0.95 | 0.99 | 4.31 K (2.00%) | 2.14 K (1.01%) | 1.13 K (0.50%) | 0.16 K (0.07%) | -10.64 K (-4.94%) | -5.15 K (-2.43%) | 10.15 K (4.72%) | 5.20 K (2.44%) |
| $H_2O$ | 0.82 | 0.85 | 0.54 ppmv (11.03%) | 0.36 ppmv (7.21%) | 0.19 ppmv (3.99%) | 0.03 ppmv (0.61%) | -1.29 ppmv (-26.77%) | -0.95 ppmv (-19.20%) | 1.28 ppmv (26.42%) | 0.96 ppmv (19.47%) |
| $O_3$ | 0.95 | 0.97 | 0.15 ppmv (9.93%) | 0.11 ppmv (6.79%) | 0.04 ppmv (1.79%) | < 0.01 ppmv (0.18%) | -0.33 ppmv (-21.04%) | -0.28 ppmv (-16.95%) | 0.31 ppmv (23.56%) | 0.27 ppmv (16.40%) |
| CO–UTLS | 0.68 | 0.78 | 17.52 ppbv (37.18%) | 13.04 ppbv (27.83%) | 6.79 ppbv (12.07%) | 0.56 ppbv (1.18%) | -32.55 ppbv (-72.60%) | -30.04 ppbv (-64.04%) | 39.92 ppbv (84.39%) | 30.66 ppbv (65.25%) |
| CO | 0.79 | 0.75 | 104.17 ppbv (62.19%) | 94.00 ppbv (58.52%) | 14.40 ppbv (12.92%) | 12.13 ppbv (7.59%) | -252.65 ppbv (-148.00%) | -257.86 ppbv (-156.18%) | 247.13 ppbv (139.53%) | 238.02 ppbv (134.42%) |
| $SO_2$ (January 2022) | 0.26 | 0.56 | 8.03 ppbv (923.39%) | 5.44 ppbv (610.92%) | 1.44 ppbv (115.73%) | 0.08 ppbv (9.16%) | -18.47 ppbv (355.89%) | -13.18 ppbv (289.23%) | 19.04 ppbv (-244.61%) | 12.83 ppbv (-238.91%) |
| $SO_2$ (May 2022) | 0.20 | 0.27 | 7.67 ppbv (1029.03%) | 6.01 ppbv (778.25%) | 1.52 ppbv (146.28%) | 0.23 ppbv (32.51%) | -17.36 ppbv (87.57%) | -14.00 ppbv (147.09%) | 17.95 ppbv (112.68%) | 14.18 ppbv (-157.35%) |
| $HNO_3$ | 0.73 | 0.72 | 0.79 ppbv (49.38%) | 0.58 ppbv (39.12%) | 0.14 ppbv (6.19%) | 0.12 ppbv (13.17%) | -1.96 ppbv (-119.60%) | -1.49 ppbv (-105.40%) | 1.66 ppbv (110.05%) | 1.03 ppbv (53.57%) |
| $N_2O$ | 0.96 | 0.94 | 26.53 ppbv (46.92%) | 8.85 ppbv (44.84%) | 22.29 ppbv (34.33%) | 0.50 ppbv (1.83%) | -16.47 ppbv (-76.91%) | -20.75 ppbv (-102.82%) | 43.84 ppbv (58.44%) | 20.76 ppbv (105.66%) |

*Author contributions.* FW, NJL, LFM, WGR, MJS, AL, and MLS have shaped the concept of this study and refined the approach during extensive discussions. FW, PAW, and WHD implemented the ANN approach into the current NRT algorithm chain. FW, LFM, and SNT carried out the data analysis and prepared the figures for the manuscript. FW wrote the initial draft of the manuscript, which was subsequently refined by all authors.

*Competing interests.* The authors declare that they have no conflict of interest.

*Acknowledgements.* ©2023. California Institute of Technology. Government sponsorship acknowledged. The research was carried out at the Jet Propulsion Laboratory, California Institute of Technology, under a contract with the National Aeronautics and Space Administration (80NM0018D0004).

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
