# Peer review of "Applying machine learning to improve the near-real-time products of the Aura Microwave Limb Sounder"

_EGUsphere, 2023_

## Author Comment (AC1)

We'd like to thank the editor for handling our manuscript, as well as reviewer #1 for reading our manuscript and providing numerous helpful suggestions for improvement.

We have carefully read through all the comments and questions and revised the manuscript accordingly. Please find our point-by-point response to reviewer #1 below. Here, the reviewer's general and specific questions/comments are formatted to be left-aligned text in bold font. Our responses are indented and formatted in regular font.

Here is a summary of the major changes in the revised manuscript:
1) Table 2 reports ANN performance metrics for both the validation and independent test data set.
2) We added additional information on the ranges for each hyperparameter and computational costs to section 3.
3) We added explanations on why the temperature ANN model appears to be more complex than other models.
4) 2) Tables 2 and 3 report the respective ANN performance metrics (RMSD, bias, and percentile differences) for each species in both their natural units (K, ppmv, ppbv), as well as percentages.
5) We added a subsection on data quality assessment to section 3.
6) We discuss areas in the global maps, where the ANN-NRT algorithm exhibits clear underestimations.

**The authors present a near real-time processor of Aura/MLS observations using a supervised neural network. The manuscript is easy to follow and shows that the processor has very good performance, very close to the operational processor. The new method presents a significant improvement compared to the previous near real-time processor based on a simplified optimal estimation method. I recommend the manuscript for publication, but I have minor comments that could be clarified by the authors.**

**General comments**

**1) I am impressed by the results overall, and more particularly with the ability of the model to capture the increase in H2O induced by the volcanic eruption, though the statistical weight of such events in the training dataset should be low. This illustrates the high potential of the model to capture special disturbances that occur over a restricted spatio-temporal range. However, I found that such abnormal conditions are not sufficiently discussed in the manuscript. Indeed, these are scientifically the most interesting cases but have a low impact on the overall statistical evaluation. For example, in Figure 4b, the increase in H2O at 100 hPa over India and part of Southeast Asia is clearly underestimated with ANN-NRT. This should be discussed in the manuscript and the authors should mention if they have found other cases where significant discrepancies were seen.**

These maps were originally thought of as simple examples. However, the reviewer is correct that we mainly focused on regions were the ANN-NRT performed well compared to the OE-NRT algorithm. We agree that it is only fair to point out areas where the ANN underperforms. However, we need to emphasize that these maps are generated from MLS observations sampled on a single day, which requires an area-weighted interpolation of the MLS orbit track. Also note that the discrete color bar can exaggerate discrepancies.

Regarding the $H_2O$ at 100 hPa example (Fig. 4b), the underestimations over India and Southeast Asia on that day are on the order of 0.5 ppmv and the OE-NRT algorithm seems to perform a little bit better. However, if we look at maps from two other days in that same week, shown in Figure 1 of this reply, we can see that ANN-NRT clearly outperforms OE-NRT in this region (as well as over Central America). While better than OE-NRT, the ANN again seems to underestimate the L2 results. Note that this is also indicated in Fig. 2h of the manuscript, where $H_2O > 5$ ppmv seem to be underestimated during that month. These apparent systematic departures of ANN $H_2O$ from the L2 training set in the presence of strong convection warrant further investigation (although it will be hard to improve the respective ANN model, due to the statistical nature of machine learning approaches).

We added some additional discussion to the revised manuscript. Here, we emphasize regions where the ANN-NRT shows some larger discrepancies to the L2 results and mention possible reasons.

[Figure]

*Fig. 2: Comparisons of L2 O₃ retrievals and ANN-NRT predictions at 100.00 hPa and 21.54 hPa.*

First, we added this to the H₂O discussion:

"A notable exception is the area of increased H₂O over India and parts of Southeast Asia, where the ANN-NRT underestimates the L2-retrieved concentrations. This region is characterized by strong and deep convection during the monsoon months that affects the sampled radiance profiles and may introduce uncertainties into the ANN model predictions. Maps of 100.00 hPa-H₂O concentrations on other days during that week indicate that slight underestimations persist in this area; however, the ANN-NRT predictions generally are much closer to the L2 results than are the OE-NRT retrievals."

We also highlight an area with pronounced O₃ underestimations:

"The only obvious difference is the area of low concentrations over Antarctica, which is completely missed by the OE-NRT algorithm and is overestimated (in area) by ANN-NRT. Note that profiles sampled in this region are affected by radiances that are reflected by the surface (see Fig. 7d in Werner et al., 2021 and the relevant discussion), which might impact the reliability of the ANN predictions."

Finally, we added this part to the conclusions:

"Global maps of predicted H₂O and O₃ concentrations indicate that model performance may be affected by the presence of strong, deep convection, as well as by strong surface reflections over Antarctica. While the respective predictions agree better with the L2 retrievals compared to the OE-NRT results, more analysis is needed to explore potential improvements to the ANN setups."

Such improvements might be achieved by increasing the sample importance for cloudy profiles (i.e., telling the model to emphasize these profiles during training) or by adding additional features that indicate cloudiness.

**2) More generally, the authors do not show results for the whole test dataset (5% of 17 years corresponds to almost 1 year), in particular winter time which is strongly disturbed in the northern hemisphere. Is there a seasonal pattern in the results? Authors should clarify why the test data are well suited for describing the capability of the model and the limitations of such a choice (that could further be investigated in future studies). For instance, I would personally have used 2 entire years with very different conditions (e.g., SSW strength or QBO phase) to test the models.**

We should have been clearer about the purpose of the independent test data set. The examples shown in the results section were not drawn from the test data set, but instead are new predictions made after each model was finalized. Note that the temperature model was trained on data sampled between 01/01/2005 and 05/31/2021. Meanwhile, Figs. 2–3 show comparisons between L2 retrievals and ANN predictions for July 2021.

The purpose of the independent test data set, and the validation data set to a certain extent, is indeed to evaluate model performance and test the ability of the model to generalize. For the temperature model, we have about 5 years and 1 year worth of profiles in the validation and test data set, i.e., about 4.8 and 1 million profiles. However, they do not comprise a continuous 5-year period or a single year of observations. Instead, these profiles are picked randomly from the full distribution and therefore cover all years, seasons, and geographical regions. If there is a close agreement between the performance metrics for the validation and test data set the model is able to generalize well for previously unseen data. Large discrepancies indicate poor model performance and unreliable predictions.

The example data in Figs. 2–4, as well as the figures in the appendix, are simply examples to illustrate performance going forward, i.e., after the models have been trained and evaluated. For example, it would not be possible to create maps like those in Fig. 4 of the manuscript from the test data set alone, because (statistically) only 5% of profiles of each individual day (~175) are part of the test data set (i.e., >5200 profiles from each month since 01/01/2005). Similarly, the analysis in Fig. 5 is based on a fixed independent data set. They are not used for model evaluation, although all predictions from now on could technically be considered an extension of the original test data set (i.e., an ever-growing amount of previously unseen profiles).

Note that we are constantly monitoring ANN-NRT performance. Fig. 2 of this reply shows an example of L2 $O_3$ retrievals vs ANN-NRT predictions from 04/09/2023 for two pressure levels. Similar to the metrics in the test data set, correlation coefficients are high with $R>0.99$ and very low biases <0.01 ppmv.

[Figure]

*Fig. 2: Comparisons of L2 O₃ retrievals and ANN-NRT predictions at 100.00 hPa and 21.54 hPa.*

We made a several changes to the revised manuscript to make these points clearer:
1)  We include the metrics for the validation data set in Table 2 and directly contrast them with the metrics for the test data set. They are very similar.
2)  We added the following description to the manuscript text:
    "These scores were derived by comparing the ANN-NRT predictions with the respective MLS L2 results for all MAFs in both the validation and an independent test data set. The distinction between the two is important. Following the discussion in Ripley (1996) and Russel and Norvig (2009), the validation data is used for hyperparameter tuning and to prevent overfitting during model training. To truly evaluate the performance of a trained model, a completely independent test data set is necessary. However, the performance scores for the validation and test data set should be similar and large discrepancies are an indication that the trained model does not generalize well (i.e., the model performs worse for previously unseen data). Note that of the ~ 3500 daily profiles MLS observed since 01/01/2005, ~ 875 and ~175 randomly selected samples are included in the validation and test data set, respectively."
3)  We've added the following statement at the beginning of section 4:
    "This section presents comparisons between MLS L2 profile retrievals and the respective OE-NRT and ANN-NRT predictions. These observations were made after the respective ANN-models were developed, trained, and evaluated and serve as examples of model performance going forward."

**3) Regarding the vertical resolution of profiles predicted with ANN-NRT. This issue is not addressed in the manuscript and could be clarified. If I understand the NN setting correctly, the vertical resolution of the predicted profile is the same as that of the level 2 operational product (here I am referring to the resolution derived from the operational averaging kernels and not the retrieval levels spacing). Am I right? This could be clarified.**

ANN-NRT retrievals are trained to duplicate the L2 operational OE retrievals, and thus have vertical and horizontal resolution no better than that inherent in the OE retrievals' averaging kernels. However, the production OE retrieval uses multiple, overlapping scans of the atmosphere to "tomographically" retrieve a set of adjacent profiles, while the ANN-NRT relies upon radiances only from the nearest radiometric scan of the atmosphere to retrieve a given profile. This difference between 2D and 1D radiance inputs would be expected to have significant impact on horizontal (along-track) resolution and more subtle impacts on vertical resolution, but as ANN-NRT retrievals do not produce averaging kernels, it is difficult to make quantitative comparisons. This is a topic for further research beyond the scope of this paper.

The ANN-NRT models perform a mapping between (1) The MLS L1B brightness temperatures (sampled at 125 minor frames, or scan levels) and the operational L2 data products at their respective 37 or 55 retrieval levels (depending on the species). The models do not approximate any parts of the forward model or retrieval algorithm.

In other words, in the training phase, the temperature ANN learns the relationship between MLS brightness temperatures sampled in different bands/channels/minor frames and the operationally retrieved $T$ at 100 hPa, 82.5 hPa, and 68.1 hPa, as well as the respective retrieval levels below and above. In the prediction phase it subsequently provides an estimate of $T$ at these exact levels, thus providing an estimate of the eventual operational L2 profile retrieval.

We've added a short summary at the beginning of section 3 of the revised manuscript: "This section described the theory, training process, settings, performance evaluation, and data quality assessment of the updated, ANN-based NRT algorithm. The goal is to train ANN models on all valid MLS L2 standard product retrievals over 01/01/2005–04/30/2022 and their associated, nearest brightness temperature profiles. Since the MLS L2 standard products are used as labels (i.e., ``truth") during training, the best-case output of each ANN is a computationally-inexpensive, high-fidelity preview of the L2 profiles."

We've also added the following clarification to section 3.1:
"Here, the labels are values from individual profiles of a specific MLS retrieved L2 atmospheric constituent. Therefore, the size of $k$ is determined by the number of retrieval levels of the respective MLS L2 product."

**4) For low SNR cases, the authors mentioned that the NN tends to smooth the noise compared to the operational product. Is this effect could be related to a degradation of the vertical resolution similar to the regularization effect in the OE method?**

"Smoothing the noise" can be thought as more of a symptom than anything the ANNs actively do. What actually happens is that the models fail to establish a successful mapping between the features and labels, i.e., the model cannot determine any meaningful relationship between the input and output.

One can test this easily with a few simple examples. In a first test, we trained a model to predict a sine curve pattern, ie., the features are angles between 0 to 360 degrees and the labels are the sine of the features. Note that, since this is just a demonstration, we did not tune any of the hyperparameters and instead used some default settings of one hidden layer, 1 neuron, a "relu" activation function, and L2 regularization with a parameter of 5e-4; the split between training, validation, and test data is 90/10/10%.

Figure 1a of this reply show the results of this test. The test data (orange dots) nicely follow the original sine curve (blue dots) and the correlation coefficient is 1.00. If we set all input features to 0, the model will fail to establish a successful mapping between the features and labels. This is illustrated in Fig. 1b of this reply. Here, the model basically predicts 0 for all angles, i.e., the average value. Some slight deviations from 0 can be observed occasionally, which can likely be attributed to (i) insufficient regularization in the model, and/or (ii) imbalanced training and test data (i.e., they draw from slightly different distributions).

A second experiment simulates conditions closer to noisy MLS L2 retrievals, shown in Fig. 1c of this reply. Here, the features are again values between 0 and 360 degrees, but we set the labels to random values between 0 to 1, with an average of 0.5. Again, the model will fail to establish a successful mapping between features and labels and will simply predict the average value at all times. This is shown in Fig. 1d of this reply.

Note that this behavior is similar to other machine learning architectures, like GBDT, where the model attempts to predict residuals from an average value.

[Figure]

*Fig. 3: Demonstration of ANN predictions for ill-defined problems.*

We slightly tweaked a statement in the abstract:
"…, where the ANN models fail to establish a functional relationship and tend to predict zero."

**Specific comments**

**Line 87: "n" is already used to define the number of input features. It would be clearer if another letter is used for the number of neurons per hidden layer.**

Thanks for noticing. We switched the index to "j" (a common letter to describe an index) and the total number to a capital "J", both in the manuscript text and in Table 1.

**Line 93: is the levels of the predicted profile the same as the number of levels of the operational product?**

This is correct. The labels of each ANN model are the respective operational retrieval products. Therefore, the predicted profiles exist at the exact same vertical levels as the MLS L2 products.

We've added a short summary at the beginning of section 3 of the revised manuscript: "This section described the theory, training process, settings, performance evaluation, and data quality assessment of the updated, ANN-based NRT algorithm. The goal is to train ANN models on all valid MLS L2 standard product retrievals over 01/01/2005– 04/30/2022 and their associated, nearest brightness temperature profiles. Since the MLS L2 standard products are used as labels (i.e., ``truth") during training, the best-case output of each ANN is a computationally-inexpensive, high-fidelity preview of the L2 profiles."

We've also added the following clarification to section 3.1:
"Here, the labels are values from individual profiles of a specific MLS retrieved L2 atmospheric constituent. Therefore, the size of $k$ is determined by the number of retrieval levels of the respective MLS L2 product."

**Table 1: I understand that the hyperparameters are defined by a set of tests but the differences between the models could be discussed. Why the number of hidden neurons is much smaller for the H2O model than for T and O3? Why is the tanh activation preferred over Relu for some species? (It is considered that Relu make the training more efficient)**

These discrepancies can be explained by the following reasons:
1) Development on the ANN-NRT models started because we were unsatisfied with the performance of the previous OE-NRT temperature results. Therefore, we initially only intended to replace the temperature product and to continue using OE-NRT for all other species. As a result, we almost overengineered that specific model and did not mind the immense computational costs associated with almost >5,000 neurons per layer. We also were content with increasing the mini-batch size to 8192, even though this required a significant amount of memory. We only cared about developing the very best model possible.
2) We made a mistake in Table 1; the $O_3$ model only has 400 neurons (as well as a "tanh" activation function).
3) Regarding the number of neurons: we varied those between 100 and a predefined maximum, in increments of 100. We set that maximum to $\frac{2}{3} \cdot$ (the number of features

+ the number of labels), which is a widely-used (somewhat empirical) threshold. Increasing the number of neurons after that point usually makes very little sense; our experience confirms these findings.

Frankly, neither the large number of neurons or the large mini-batch size for the temperature model are necessary. In fact, as long as the number of neurons is ≥400 per layer, the overall performance metrics change very little (e.g., $\Delta R < 0.01$). Once we decided to also train models for the other NRT species, we decided to keep the mini-batch size lower to ease the computational costs regarding the amount of memory, as we found little to no improvement for the performance metrics. However, we decided to keep the already trained temperature model the way it was.

Regarding the use of the "tanh" vs "relu" activation functions: We found that for almost all models the performance was determined by the combination of activation function and normalization. Apart from the $O_3$ model, the use of "relu" only produced higher performance scores when combined with Gaussian noise layers. Whenever L2 regularization was associated with higher performance (or no regularization was preferrable, like for the CO models) it was in combination with "tanh" layers. Note that we are not saying this is a universal characteristic, but something unique to the MLS NRT setup. Similar to our findings for the temperature model: differences in the performance metrics between the different model setups were very small as long as we had a sufficient number of neurons per hidden layer. However, the reviewer is correct: the models with "relu" activation functions converged a lot faster than the "tanh" models. This is one of the large benefits of the "relu" activation function. Of course, there are also some disadvantages, like the fact that neurons with negative values get eliminated.

We added the considered ranges of each hyperparameter to section 3.1:

"We considered the following ranges and settings: $J_{HL}$ = [1, 2], $J_N$ = [100, 200, · · ·, 2/3·$(n+k)$] per hidden layer, AF=["relu", "tanh"], LRP=[n/a, 1e−6, 5e−6, 1e−5, · · ·, 1e−1], GNS=[n/a, 1e−3, 5e−3, 1e−2, · · ·, 1], and MBS=[32, 64, · · ·, 8192].

We also added information on the computational costs of the training procedure: "The computational costs associated with the training procedure of each ANN-NRT model, while dependent on the respective hyperparameters and size of the $m$ x $n$ input matrix, are generally as follows: it takes about one month to develop and train each ANN, using 12 CPUs and requiring ~ 100 GB of memory."

Finally, we added an explanation on why the temperature model is so much more complex:
"Note that the model setups for $T$, CO, and $SO_2$ differ from those of the other species. The $T$ model is considerably more complex with comparatively high values of $J_{HL}$=5,078 and MBS=8,192. The ANN-based estimator for temperature was developed before those for the other products, with less regard for computational cost than was present in the subsequent development. The computationally more expensive temperature model is

"overbuilt", but had already been trained so was used in this version of the NRT products."

**Line188/Table 3: Are the scores calculated for the same periods as Figure 3?**

Yes, these metrics were calculated for the same time period as in Fig. 2, 3, A1, and A2. We clarified this in the revised manuscript:
"A summary of average performance metrics is given in Table 3, derived for the same time period as is used in Figs. 2–3 and Figs. A1–A2. Specifically, the presented metrics are: $R$, the average absolute RMSD, and average absolute bias for each species and the two NRT algorithms, as well as the averages of the 1st and 99th percentile of the differences to L2 (as a proxy for the minimum and maximum deviations)."

**Line207: "Here the ANN … , and the results are close to L2 data": there is a clear underestimation of the H2O vmr over india and East-Asia. This issue could be mentioned and what could be the reason?**

- Yes, these are random.

**Line 219/Line 241: Would it be possible to complete a small training dataset with simulated data?**

Yes, this could be done. Testing performance from a model that was trained on simulated data would be an interesting analysis. It would prove the feasibility of providing NRT products for a completely new instrument, for which no previous data record exists. Since machine learning approaches are statistical in nature, using a wide array of synthetic composition profiles and radiance data should in theory provide reliable predictions. Such an approach is not dissimilar to calculating look-up tables of synthetic observations for a wide range of viewing geometries and cloud variables in MODIS-like cloud property retrievals. As long as the radiances accurately describe the actual (noisy) observations and the set of composition profiles cover a wide array of possible atmospheric states, that approach should yield reliable results. Again, a retrieval approach based on look-up tables is very similar.

We think that such an analysis goes far beyond the scope of our paper and would be best suited for a separate study. For the MLS NRT retrieval we fortunately did not require synthetic data sets due to the large MLS data record.

Some preliminary thoughts on such a new study are:
1) The synthetic profiles need to be representative of actually observed profiles, and should cover as much of the full dynamic range as possible. We would expect to need an order of magnitude of 100,000 profiles to develop a reliable model.
2) For each of these profiles we need to simulate the relevant MLS radiance observations, which is computationally expensive.
3) ANNs might not be the ideal machine learning architecture for such an application. They tend to learn a specific problem very well, such as measurement uncertainties

and idiosyncrasies in the applied forward model and inversion algorithms, which result in uncertainties in the retrieved composition profiles. Synthetic data might look just different enough compared to actual measurements/retrievals that the ANN predictions become unreliable. Other architectures, such as Gaussian Process Models or Gradient Boosted Decision Trees, are more robust with regard to noisy data.

We think this topic is well worth exploring in a separate study, but that it requires a lot of additional efforts and considerations.

However, we ran a small test to, at the very least, confirm the feasibility of such an approach. Instead of creating a large set of possible atmospheric states and running a forward model on each to create synthetic MLS radiances, we used simulated radiances for day 51 in 1996 as input for our ANN-NRT temperature model. That data set is part of our testing procedure for the MLS retrieval algorithm. Note that the ANN-NRT models were trained on the relationship between a set of noisy MLS radiances and noisy MLS L2 retrievals. Applying these models on noise-free radiances and climatological temperature profiles introduces considerable uncertainties.

The results are shown in Fig. 4 of this reply. Panels a and b show scatter plots of predicted vs modelled temperatures at 100.00 hPa and 21.54 hPa, respectively. While model performance is worse compared to our analysis for actually observed MLS radiances and retrieved temperature profiles, it still performs reasonably well. Correlation coefficients are 0.95 (100.00 hPa) and 0.93 (21.54 hPa). The RMSD>2 K is in the range of the results in table 3 of the manuscript. These metrics are also worse than the ones based upon a single year of MLS observations (see Fig. 5 of the manuscript).

[Figure]

*Fig. 4: Model performance for simulated temperature profiles and radiances.*

**Line 244: I don't understand the sentence "The previous version…". Do the authors mean: The previous version of MLS NRT data products (OE-NRT, Lambert et al., 2022) is replaced with predictions from an artificial neural network (ANN).**

This is indeed confusing. The ANN approach was developed and implemented in phases, starting with the temperature ANN model and only later extended to cover all other NRT species as well. An ANN-based model has been used operationally for NRT temperature

since the end of 2021, as documented in the previous version of the MLS NRT user guide.

However, we don't think this distinction is necessary and will only confuse potential readers. We therefore simplified the first paragraph of the conclusions in the revised manuscript to:
"The previous version of MLS NRT data products (OE-NRT) is replaced with predictions from an artificial neural network (ANN). This manuscript describes the setup and evaluation of ANN models for all MLS NRT species. Starting in January 2023, all MLS NRT data products are based on this new approach (ANN-NRT)."

---

## Author Comment (AC2)

We'd like to thank the editor for handling our manuscript, as well as reviewer #2 for reading our manuscript and providing numerous helpful suggestions for improvement.

We have carefully read through all the comments and questions and revised the manuscript accordingly. Please find our point-by-point response to reviewer #2 below. Here, the reviewer's general and specific questions/comments are formatted to be left-aligned text in bold font. Our responses are indented and formatted in regular font.

Here is a summary of the major changes in the revised manuscript:
1) Table 2 reports ANN performance metrics for both the validation and independent test data set.
2) We added additional information on the ranges for each hyperparameter and computational costs to section 3.
3) We added explanations on why the temperature ANN model appears to be more complex than other models.
4) 2) Tables 2 and 3 report the respective ANN performance metrics (RMSD, bias, and percentile differences) for each species in both their natural units (K, ppmv, ppbv), as well as percentages.
5) We added a subsection on data quality assessment to section 3.
6) We discuss areas in the global maps, where the ANN-NRT algorithm exhibits clear underestimations.

**General comments**

**This paper presents new near real-time products of the Aura Microwave Limb Sounder (MLS) using artificial neural networks (ANN-NRT). The ANN-NRT show good performance and demonstrates the potential of applying machine learning to generate NRT products. The paper is clearly written and the study is well explained. I recommend the manuscript for publication, but I have some minor comments.**

**(1) Global maps show ANN-NRT is better than OE-NRT, but more discussion should be given to the special area of that ANN overestimates or underestimates.**

These maps were originally thought of as simple examples. However, the reviewer is correct that we mainly focused on regions were the ANN-NRT performed well compared to the OE-NRT algorithm. We agree that it is only fair to point out areas where the ANN underperforms. However, we need to emphasize that these maps are generated from MLS observations sampled on a single day, which requires an area-weighted interpolation of the MLS orbit track. Also note that the discrete color bar can exaggerate discrepancies.

We added some additional discussion to the revised manuscript. Here, we emphasize regions where the ANN-NRT shows some larger discrepancies to the L2 results and mention possible reasons. First, we added this to the $H_2O$ discussion:

"A notable exception is the area of increased $H_2O$ over India and parts of Southeast Asia, where the ANN-NRT underestimates the L2-retrieved concentrations. This region is characterized by strong and deep convection during the monsoon months that affects the sampled radiance profiles and may introduce uncertainties into the ANN model predictions. Maps of 100.00 hPa-$H_2O$ concentrations on other days during that week indicate that slight underestimations persist in this area; however, the ANN-NRT predictions generally are much closer to the L2 results than are the OE-NRT retrievals."

We also highlight an area with pronounced $O_3$ underestimations:
"The only obvious difference is the area of low concentrations over Antarctica, which is completely missed by the OE-NRT algorithm and is overestimated (in area) by ANN-NRT. Note that profiles sampled in this region are affected by radiances that are reflected by the surface (see Fig. 7d in Werner et al., 2021 and the relevant discussion), which might impact the reliability of the ANN predictions."

Finally, we added this part to the conclusions:
"Global maps of predicted $H_2O$ and $O_3$ concentrations indicate that model performance may be affected by the presence of strong, deep convection, as well as by strong surface reflections over Antarctica. While the respective predictions agree better with the L2 retrievals compared to the OE-NRT results, more analysis is needed to explore potential improvements to the ANN setups."

Such improvements might be achieved by increasing the sample importance for cloudy profiles (i.e., telling the model to emphasize these profiles during training) or by adding additional features that indicate cloudiness.

**(2) For performance evaluation of T model, I think it is more intuitive to use unit K rather than relative values. At least it should be described in the paper.**

Tables 2 and 3 in the revised manuscript now summarize both Kelvin/ppmv/ppbv, as well as percentages. This not only provides more intuitive numbers for the temperature model, but also puts some of the large percentages for $SO_2$, $HNO_3$, and $N_2O$ into perspective (i.e., there are very low concentrations at certain levels).

**Specific comments**
**Line 96: I know brightness temperatures sampled over 2005–2022 are very large. However, it is better to describe the exact amount of input features for training, validation, and test data.**

At that point in the manuscript, we wanted to give a very general overview of the theory and necessary steps to setup and train ANN models. Moreover, the exact number of samples varies from species to species due to the (i) differently sized data records, and (ii) number of successful MLS level 2 profile retrievals.

However, we agree that this is an important fact to cover in the revised manuscript. Therefore, we added the respective number of samples to table 1 and changed the relevant sentence in the manuscript text to: "It also provides details on the features that make up the input matrix for each ANN-NRT model, namely the start and end dates that define the training data record for each model, the number of total samples in that data record (determined by the number of successful profile retrievals), and the respective MLS bands, channels, and MIFs."

**Table 1: The number of neurons of T and O3 are much larger than other products, is it necessary? Why choose so many neurons instead of adding hidden layers? The MBS of T (i.e. 8192) is much larger than the others (i.e. 32), it should be discussed.**

These discrepancies can be explained by the following reasons:
1) Development on the ANN-NRT models started because we were unsatisfied with the performance of the previous OE-NRT temperature results. Therefore, we initially only intended to replace the temperature product and to continue using OE-NRT for all other species. As a result, we almost overengineered that specific model and did not mind the immense computational costs associated with almost >5,000 neurons per layer. We also were content with increasing the mini-batch size to 8192, even though this required a significant amount of memory. We only cared about developing the very best model possible.
2) We made a mistake in Table 1; the $O_3$ model only has 400 neurons.
3) Regarding the number of neurons: we varied those between 100 and a predefined maximum, in increments of 100. We set that maximum to $\frac{2}{3} \cdot$ (the number of features

+ the number of labels), which is a widely-used (somewhat empirical) threshold. Increasing the number of neurons after that point usually makes very little sense; our experience confirms these findings.

Frankly, neither the large number of neurons or the large mini-batch size for the temperature model are necessary. In fact, as long as the number of neurons is ≥400 per layer, the overall performance metrics change very little (e.g., $\Delta R < 0.01$). Once we decided to also train models for the other NRT species, we decided to keep the mini-batch size lower to ease the computational costs regarding the amount of memory, as we found little to no improvement for the performance metrics. However, we decided to keep the already trained temperature model the way it was.

We added the considered ranges of each hyperparameter to section 3.1:

"We considered the following ranges and settings: $J_{HL}$ = [1, 2], $J_N$ = [100, 200, · · ·, 2/3·$(n+k)$] per hidden layer, AF=["relu", "tanh"], LRP=[n/a, 1e−6, 5e−6, 1e−5, · · ·, 1e−1], GNS=[n/a, 1e−3, 5e−3, 1e−2, · · ·, 1], and MBS=[32, 64, · · ·, 8192].

We also added information on the computational costs of the training procedure:
"The computational costs associated with the training procedure of each ANN-NRT model, while dependent on the respective hyperparameters and size of the $m$ x $n$ input matrix, are generally as follows: it takes about one month to develop and train each ANN, using 12 CPUs and requiring ~ 100 GB of memory."

Finally, we added an explanation on why the temperature model is so much more complex:
"Note that the model setups for $T$, CO, and SO$_2$ differ from those of the other species. The $T$ model is considerably more complex with comparatively high values of $J_{HL}$=5,078 and MBS=8,192. The ANN-based estimator for temperature was developed before those for the other products, with less regard for computational cost than was present in the subsequent development. The computationally more expensive temperature model is "overbuilt", but had already been trained so was used in this version of the NRT products."

**Line 189: The SO2 statistics in Table 3 are based on the observations which were also included in training data set. So, the comparison of OE and ANN doesn't make much sense. Is there no other data for comparison?**

We wanted to present statistics for the data covered in Figs. 2-3 and to present model performance for enhanced concentrations due to volcanic activity. However, we agree that the evaluation of the SO$_2$ model performance is problematic due to the inclusion of trained data. We acknowledge that fact in the manuscript when we say:
"Of special note is the ANN-NRT setup for sulphur dioxide (SO2). Volcanic eruptions are the primary source of stratospheric SO$_2$. As a result, we decided to train the SO$_2$ ANN model on MLS observations around major volcanic eruptions, namely those of Kasatochi, Calbuco, Sarychev, Nabro, Raikoke, and Hunga Tonga-Hunga Ha'apai (e.g., Pumphrey

et al., 2015; Millán et al., 2022). While ANN-NRT performs well in reproducing elevated $SO_2$ concentrations associated with the Hunga Tonga-Hunga Ha'apai eruption, the training data is limited and the model may suffer from overfitting (i.e., learning specific characteristics of known eruptions well to the detriment of generalization)."

We agree that adding more information about model performance for actually unseen data are necessary. Therefore, we added performance metrics for predictions in May 2022 to table 3, as well as the following sentence in the manuscript text:
"Note that two sets of $SO_2$ statistics are shown: one set based on MLS observations in January 2022, which are affected by the Hunga Tonga-Hunga Ha'apai volcanic eruption and were included in training data set, and a second set based on samples in May 2022 with no volcanic influence."

We also developed a second $SO_2$ ANN-NRT model, where the training data set is based on all MLS observations over 01/01/2005–04/30/2022. We included performance metrics for the test data set in table 2, as well as the following discussion in the revised manuscript:
"As mentioned in section 2, stratospheric L2 retrievals in the absence of elevated levels of $SO_2$ can be considered noise, and comparisons between L2 and ANN-NRT results are difficult ($R<0.26$ and bias $>11\%$). If the training data set is increased to include all MLS retrievals between 01/01/2005 and 04/30/2022, instead of just focusing on periods of volcanic activity, the associated correlation coefficients and biases slightly improve to 0.37 and $<7\%$, indicating a better ability to predict noise. However, further analysis indicates that this model performs slightly worse for profiles containing elevated $SO_2$ concentrations; correlation coefficients for such profiles in the test data set are decreased by about 0.05 ($R=0.52$ compared to $R=0.57$), while the RMSD increases by about 0.31 ppbv (5.72 ppbv compared to 5.41 ppbv). Since the main objective of the $SO_2$ NRT is to detect volcanic activity, we decided to employ the model trained on the reduced (volcanic only) data set."

**Line 237: All metrics get better with the increasing data except the absolute bias in Fig. 5(c), it should be discussed.**

Unfortunately, after closely analyzing the different predictions and metrics, we frankly do not have a good explanation on why the bias does not decrease with increasing training data set size.

One possible explanation is that the observed biases between predictions are very small, especially compared to other species. Looking at the new table 3, the mean bias for the full data set is 0.16 K ($<0.1\%$). The bias range for the smaller training data sets is 0.11-0.13 K, which is very similar. Keep in mind that these are absolute biases, so the true average is even smaller at 0.05–0.06 K. At this point, the biases for all ANN-NRT models are close to being negligible and increasing the size of the training data set does not further reduce the bias.

Note that the average of the 99[th] percentile of the difference between the L2 retrievals and ANN predictions decreases with an increase in training data size; illustrated in Fig. 1 of this reply. We find similar results for the 1[st] percentile of the difference.

[Figure]

*Fig. 1: (a) Correlation coefficient as a function of size of the training data set. (b) Similar to (a), but for the average of the 99[th] percentile of the difference between L2 and predicted temperatures. (c) Similar to (b), but shown as a percentage.*

We unfortunately do not have another explanation; all predictions look very similar. Even though this lack of an explanation is rather unsatisfactory, we added some extra discussion about the small biases and the 99[th] percentile differences to the revised manuscript:

"A very small increase in the averaged absolute biases for the *T* models is observed. However, these absolute biases are in the range of 0.11-0.16 K (0.05-0.06 K if both positive and negative biases are averaged) and can be considered negligible. Note that similar analysis for the 1[st] and 99[th] percentile of the difference between MLS L2 retrievals and each ANN-NRT model prediction shows a monotonically decreasing behavior with increasing training data size."

---

## Author Comment (AC3)

We'd like to thank the editor for handling our manuscript, as well as reviewer #3 for reading our manuscript and providing numerous helpful suggestions for improvement.

We have carefully read through all the comments and questions and revised the manuscript accordingly. Please find our point-by-point response to reviewer #3 below. Here, the reviewer's general and specific questions/comments are formatted to be left-aligned text in bold font. Our responses are indented and formatted in regular font.

Here is a summary of the major changes in the revised manuscript:
1) Table 2 reports ANN performance metrics for both the validation and independent test data set.
2) We added additional information on the ranges for each hyperparameter and computational costs to section 3.
3) We added explanations on why the temperature ANN model appears to be more complex than other models.
4) 2) Tables 2 and 3 report the respective ANN performance metrics (RMSD, bias, and percentile differences) for each species in both their natural units (K, ppmv, ppbv), as well as percentages.
5) We added a subsection on data quality assessment to section 3.
6) We discuss areas in the global maps, where the ANN-NRT algorithm exhibits clear underestimations.

**GENERAL COMMENTS**
==================

The paper describes the application of an artifical neural network (ANN) to the retrieval of trace gas profiles from the MLS instrument. ANN have been applied recently to different problems, partially with large success.

Here, the intent is to replace a primarily fast but comparatively inaccurate near-real-time retrieval with something both faster and more accurate. The presented results indicate that the approach has succeeded on both ends.

The study is on the point, well described, and executed. The topic fits the journal. I recommend publication.

**SPECIFIC COMMENTS**
==================

**lines 80ff: The underlying software seems to be readily available. Could the training model employed here be made available as well? This might be applicable for similar tasks and/or other limb sounders.**

> The referenced "Keras" and "Tensorflow" software packages are open source tools to set up machine learning platforms. Internally, we use specifically developed Python routines to access those open source tools and to streamline the training process. We are currently in the process of preparing a Python package that could be hosted on Github and made available to the public. However, note that these are simply wrappers to simplify access to "Keras" and "Tensorflow.

> Following the steps outlined in the Keras user manual (https://keras.io/guides/sequential_model/) to set up a feedforward neural network and using the settings summarized in Section 3.2 and Table 1 of the manuscript is all it takes to set up the exact models described in the manuscript. However, it is highly unlikely that these exact models produce reliable results for different tasks or instruments. Instead, the correct settings need to be determined individually for each application and data set; these settings are probably very different from the ones used here.

> Please reach out if you want help with setting up similar models for different tasks, we are happy to help.  (…so long as we aren't forbidden to do so by US export controls.)

**lines 130ff: A general problem with trained models is how the model copes with unexpected situations. Here, you describe how you adapted the training data set to cope with volcanic activity. How important was this for the performance and how likely is it that, e.g. the Ozone hole would have been missed?**

> This problem (performance for situations not seen before) is indeed inherent in all supervised machine-learning applications, not just for the $SO_2$ model described in the manuscript. However, the MLS $SO_2$ profile retrievals are somewhat special (compared to

the other species) as they are basically noise at all levels in the absence of volcanic activity. An example of that is presented in Fig. 1 of this reply, which shows joint histograms of the operational L2 $SO_2$ concentrations and those provided by OE-NRT, ANN-NRT trained with the reduced data set, and ANN-NRT trained with all data over 01/01/2005–04/30/2022. Joint histograms are shown for two pressure levels; data is from MLS observations in May 2022.

[Figure]

*Fig. 1: Joint histograms of L2 $SO_2$ and three NRT models for two pressure levels each.*

The ANN-NRT that was trained with all profiles performs better in predicting the noise at each pressure level than the model trained with the reduced data set. This is simply due to the fact that the former was trained on more noisy data. Both ANN-based models perform better than OE-NRT. Note the slight tilt of the ANN-based distributions in relation to the 1:1 line, which illustrates the tendency of the ANNs to predict 0 ppmv.

Both the correlation coefficient and bias for an independent test data set improve when using the ANN model that was trained on all data over 01/01/2005–04/30/2022. Average R=0.37 and average bias =5.73%, compared to 0.26 and 7.26%, respectively. However, this only illustrates that the model can predict noise better. We compared model performance for profiles with elevated values (from the independent test data set) and found that the model trained on the limited data set performs better (correlation coefficients of 0.57 vs 0.52; RMSDs of 5.41 ppbv vs 5.72 ppbv). In other words, the model trained on the full data set focuses slightly too much on the noise.

We added additional information to the revised manuscript. Table 2 now lists the performance metrics for the two different models. In the text we then motivate the use of the model trained on the limited data set:

"If the training data set is increased to include all MLS retrievals between 01/01/2005 and 04/30/2022 (named 2nd model in Table 2) rather than being restricted to periods of volcanic activity, the associated correlation coefficients and biases slightly improve to 0.37 and <7%, indicating a better ability to predict noise. However, further analysis indicates that this model performs slightly worse for profiles containing elevated $SO_2$ concentrations; correlation coefficients for such profiles in the test data set are decreased by about 0.05 ($R$=0.52 compared to $R$=0.57), while the RMSD increases by about 0.31 ppbv (5.72 ppbv compared to 5.41 ppbv). Since the main objective of the $SO_2$ NRT is to detect volcanic activity, we decided to employ the model trained on the reduced (volcanic only) data set."

**lines 235ff: This result suggests that the training data set contains a lot of redundancy, as is expected for such a large set measuring effectively the same planet all over. Do you have means to identify profiles with high influence on the training performance? And if yes, what were they?**

This would be an interesting analysis. There are several ways to determine feature importance, i.e., determine which input variables (MLS bands, channels, MIFs) are most important during the prediction. There also ways to set sample weights, i.e., a way to make sure certain profiles are more important than others.

Unfortunately, identifying individual profiles that most contributed to the training performance is not possible, at least not with our current setup. That's because during training the loss function is not calculated for individual profiles, but for a collection of profiles (called a batch). The batch size is determined by the "mini-batch size" parameter (listed in table 1 of the manuscript), which in our case is almost always 32. That means that during each training iteration, an average loss is calculated for $M$ batches (where $M$ is the total number of profiles, divided by 32). Each of the $M$ batches contains 32 randomly selected profiles. After each iteration, the ANN weights are updated based on the average loss, the input data get randomly shuffled and assigned to a new set of $M$ batches, and a new loss is calculated.

There is the possibility to set the mini-batch size to 1. However, this is not recommended as the calculated losses become very noisy, which almost certainly will prevent the model from converging. It also means that during each iteration we have to loop over every profile in the training data set, which dramatically increases the training time.

**Do you foresee a possibility to generate a synthetic set of training data for a new instrument, for which no historic data is available? How would this compare for instruments, which measure more seldomly, such as ACE-FTS. Would a year of data still be sufficient to train the retrieval?**

Machine learning approaches are statistical in nature. Using a wide array of synthetic composition profiles and radiance data should indeed provide the means to facilitate near-real-time predictions for a new instrument. Such an approach is not dissimilar to calculating look-up tables of synthetic observations for a wide range of viewing

geometries and cloud variables in MODIS-like cloud property retrievals. As long as the radiances accurately describe the actual (noisy) observations and the set of composition profiles cover a wide array of possible atmospheric states, that approach should yield reliable results. Again, a retrieval approach based on look-up tables is very similar.

We ran a small test to, at the very least, confirm the feasibility of such an approach. Instead of creating a large set of possible atmospheric states and running a forward model on each to create synthetic MLS radiances, we used simulated radiances for day 51 in 1996 as input for our ANN-NRT temperature model. That data set is part of our testing procedure for the MLS retrieval algorithm. Note that the ANN-NRT models were trained on the relationship between a set of noisy MLS radiances and noisy MLS L2 retrievals. Applying these models on noise-free radiances and climatological temperature profiles introduces considerable uncertainties.

The results are shown in Fig. 2 of this reply. Panels a and b show scatter plots of predicted vs modelled temperatures at 100.00 hPa and 21.54 hPa, respectively. While model performance is worse compared to our analysis for actually observed MLS radiances and retrieved temperature profiles, it still performs reasonably well. Correlation coefficients are 0.95 (100.00 hPa) and 0.93 (21.54 hPa). The RMSD>2 K is in the range of the results in table 3 of the manuscript. These metrics are also worse than the ones based upon a single year of MLS observations (see Fig. 5 of the manuscript).

This approach might also be preferrable for instruments with low sample frequency. ACE-FTS, for example, samples about 5,000 composition profiles per year. In our experience this is roughly an order of magnitude too low to train a reliable machine learning model. However, there are a number of data augmentation techniques (like applying Gaussian noise to the input features, as well as to the neuron output in the model) that can make the model predictions more robust even for smaller datasets.

[Figure]

Fig. 2: Model performance for simulated temperature profiles and radiances.

We performed another small test, where we tried to predict ACE-FTS $CH_4$ based only on ACE-FTS $N_2O$ concentrations, i.e., predicting the relationship shown in Fig. 1 of Minnschwaner and Manney (2014). While not the same thing as relating radiances to

composition profiles, it still gives us an idea about the impact of data set size. We compared model performance for a model that was trained on 5% (~1 year) to the performance of a model that was trained on 25% (~5 years) of data. The size of each validation data set is 2% of all ACE-FTS data up to 2022. We then compared the results for the remaining data points (i.e., test data); the results are shown in Fig. 3 of this response.

[Figure]

Fig. 3: ACE-FTS predictions of CH4.

Overall, there is not a lot of difference between the two models. Using 5 years of data increases the correlation coefficient from 0.993 to 0.994. The RMSD is the same between the two models at 0.062 ppmv. Naturally, the relationship between radiances and compound profiles is a lot more complex than the relationship between $N_2O$ and $CH_4$.

While we don't think it makes sense to add this analysis to the revised manuscript, we changed the last sentence of the conclusions to the following:
"…, which demonstrates the potential of applying machine learning to generate NRT products for other current and future mission concepts with similar sampling frequency. Alternative approaches, like training ANNs on synthetic profiles of atmospheric constituents and simulated brightness temperatures, may be needed for instruments with significantly lower sampling rates."

Reference: Minschwaner, K., Manney, G.L. Derived methane in the stratosphere and lower mesosphere from Aura Microwave Limb Sounder measurements of nitrous oxide, water vapor, and carbon monoxide. J Atmos Chem 71, 253–267 (2014). https://doi.org/10.1007/s10874-015-9299-z

**lines 244ff: Typically, level 2 products are associated with a zoo of diagnostic data from precision to resolution etc. How is the data provided by the ANN characterised?**

This is one of the disadvantages of neural networks compared to Random Forests (another popular machine learning framework): the usual implementation of neural networks does not supply any uncertainty information. However, we attempt to estimate the precision of the ANN predictions based on statistics. We also perform some basic data quality checks.

We agree that it is important to add this information to the revised manuscript. We therefore added a new subsection on data quality to section 3 of the revised manuscript. Here is a quick summary of the information:

The only data quality flag that is used going forward is the precision, which is derived as the root mean square of (i) the typical MLS L2 precisions for the given pressure level taken from the training data set, and (ii) RMSD between MLS L2 products and the predictions for the independent test data set. Negative precisions are assigned to values outside the valid pressure range, profiles in overlap regions. Data values with negative precisions should not be used. An additional data quality check assures that predictions at each pressure level are within a predefined confidence range; precisions for profiles where predictions are outside that confidence range (at any pressure level) are set to negative 1.

Note that this information is also given in the Version 5 Level-2 Near-Real-Time Data User Guide.

**lines 261ff: The speed-up of the NRT retrieval is impressive and very useful for the purpose of providing near-real-time data. How does this relate to the computational effort for training the model? Is this (over the foreseen runtime) still a net positive or does one trade in training effort for faster operational results? Does one need a super-computer/cloud service for training or is this feasible with a well-equipped work station?**

The computational costs of training the ANN-NRT models are not too crazy. The exact numbers depend on the specific model setup (number of hidden layers and neurons, mini-batch size) and the size of the input matrix (number of features and samples). Training a model on 1 year of MLS $O_3$ data, for example, requires about 60 GB of memory and takes a ~10 hours to converge when trained using 16 CPUs. The size of the data set (i.e., how many years are included) does not affect the memory requirements for the training process, as the model calculates average losses for a batch of samples; adding more data of course affects overall memory usage because the data needs to be readily available. Training the $O_3$ model on 18 years of data requires about 100 GB of memory and takes about 1 week to fully train when using 16 CPUs. This means that including the time it took to determine the best hyperparameters, each ANN-NRT model can be set up and trained in about 1 month.

A well- equipped work station is sufficient to develop and train these ANN models. Note that tree-based machine learning architectures, like Random Forests and Gradient Boosted Decision Trees, can offer similar performance at a fraction of the computational costs. These models also convergence significantly faster than ANNs.

We added the following information to the revised manuscript:
"The computational costs associated with the training procedure of each ANN-NRT model, while dependent on the respective hyperparameters and size of the $m$ x $n$ input matrix, are generally as follows: it takes about one month to develop and train each ANN, using 12 CPUs and requiring $\approx$100 GB of memory.

**lines 263ff: Are NRT retrievals the only application of the ANN model discussed here? Could this data serve as an initial guess to the OE to speed up convergence or are there reasons not to use this?**

The new ANN-based NRT predictions, as well as the previous OE-based results, could theoretically be used as an a priori guess for the operational retrieval. There are, however, a number of reasons why we have no plans of doing so:

(1) A well-defined retrieval problem should converge to the correct solution almost independent of the specific a priori profile.

(2) The retrieval uncertainty/precision would be different, as the a priori uncertainty would be different.

(3) At this point, 18 years into the MLS mission, we are avoiding massive changes to the rather complex L2 retrieval algorithm. Using the ANN predictions as a priori profiles would require a significant development and testing effort, with possibly little to no benefits (see point 1).